# Rabphilin-3A negatively regulates neuropeptide release, through its SNAP25 interaction

**Adlin Abramian[1][†], Rein I Hoogstraaten[1][†], Fiona H Murphy[1], Kathryn F McDaniel[1], Ruud F Toonen[1], Matthijs Verhage[1,2]***

[1]Department of Functional Genomics, Faculty of Exact Science, Center for Neurogenomics and Cognitive Research, Vrije Universiteit Amsterdam and Vrije Universiteit Medical Center, Amsterdam, Netherlands; [2]Department of Clinical Genetics, Center for Neurogenomics and Cognitive Research, Amsterdam Neuroscience, Vrije Universiteit Medical Center, Amsterdam, Netherlands

*For correspondence:
matthijs@cncr.vu.nl

[†]These authors contributed equally to this work

Competing interest: The authors declare that no competing interests exist.

**Abstract** Neuropeptides and neurotrophins are stored in and released from dense core vesicles (DCVs). While DCVs and synaptic vesicles (SVs) share fundamental SNARE/SM proteins for exocytosis, a detailed understanding of DCV exocytosis remains elusive. We recently identified the RAB3-RIM1 pathway to be essential for DCV, but not SV exocytosis, highlighting a significant distinction between the SV and DCV secretory pathways. Whether RIM1 is the only RAB3 effector that is essential for DCV exocytosis is currently unknown. In this study, we show that rabphilin-3A (RPH3A), a known downstream effector of RAB3A, is a negative regulator of DCV exocytosis. Using live-cell imaging at single-vesicle resolution with RPH3A deficient hippocampal mouse neurons, we show that DCV exocytosis increased threefold in the absence of RPH3A. RAB3A-binding deficient RPH3A lost its punctate distribution, but still restored DCV exocytosis to WT levels when re-expressed. SNAP25-binding deficient RPH3A did not rescue DCV exocytosis. In addition, we show that RPH3A did not travel with DCVs, but remained stationary at presynapses. RPH3A null neurons also had longer neurites, which was partly restored when ablating all regulated secretion with tetanus neurotoxin. Taken together, these results show that RPH3A negatively regulates DCV exocytosis, potentially also affecting neuron size. Furthermore, RAB3A interaction is required for the synaptic enrichment of RPH3A, but not for limiting DCV exocytosis. Instead, the interaction of RPH3A with SNAP25 is relevant for inhibiting DCV exocytosis.

## eLife assessment

This **important** study advances our understanding of the mechanisms of neuronal large dense-core vesicle (LDCV) secretion, which mediates neuropeptide and neurotrophin release. It describes a negative regulatory process involving the interaction of the Rab3-effector Rabphilin-3A with the SNARE fusion protein SNAP25, which limits LDCV secretion and neurite growth. The evidence in support of the authors' claims is generally **convincing**, but some conclusions, e.g regarding the role of Rabphilin-3A-controlled neurotrophin signaling in neurite growth, are **incompletely** supported. This study will be of interest to the fields of cell biology, cellular neuroscience, and neuroendocrinology.

## Introduction

Neuropeptides and neurotrophins are crucial neuronal signaling molecules that play diverse roles in brain development, neurogenesis, and synaptic plasticity (*Malva et al., 2012*; *Pang et al., 2004*; *Park and Poo, 2013*; *van den Pol, 2012*; *Zaben and Gray, 2013*). These neuromodulators are sorted at the trans-Golgi network and packaged into dense core vesicles (DCVs). Similar to neurotransmitter release from synaptic vesicles (SVs), neuropeptide release is calcium and activity-dependent. However, unlike SVs, DCVs can fuse at various locations in the neuron, requiring higher and more sustained stimulation frequencies (*Balkowiec and Katz, 2002*; *Hartmann et al., 2001*; *Persoon et al., 2018*). Interestingly, all SNARE/SM proteins essential for exocytosis are shared between these two secretory pathways (*Arora et al., 2017*; *Farina et al., 2015*; *Hoogstraaten et al., 2020*; *Puntman et al., 2021*; *Südhof, 2013*; *van de Bospoort et al., 2012*). However, while SV secretory pathways are well characterized, a detailed understanding of DCV exocytosis is still emerging. We recently discovered that RAB3, a protein largely dispensable for SV exocytosis (*Schlüter et al., 2006*; *Schlüter et al., 2004*), and its effector RIM1 to be essential for DCV exocytosis (*Persoon et al., 2019*). This indispensable role of the RAB3-RIM pathway signifies a main difference between SV and DCV exocytosis. Whether RIM is the only RAB3 effector that is essential for DCV exocytosis is currently unknown.

Rabphilin-3A (RPH3A) is a downstream effector of RAB3A that is highly expressed in the brain (*Schlüter et al., 1999*). RPH3A expression increases throughout development, similar to established synaptic proteins (*Baldarelli et al., 2021*; *Blake et al., 2021*; *Krupke et al., 2017*). RPH3A binds RAB3A and RAB27 and contains two lipid- and calcium-binding C2 domains (*Guillén et al., 2013*; *Tsuboi and Fukuda, 2005*). The second C2 domain (C2B) binds SNAP25 in a calcium-independent manner (*Deák et al., 2006*). Similar to RAB3 null mutants, depletion of RPH3A shows no detectable synaptic phenotype in mice under normal physiological conditions (*Schlüter et al., 1999*), and only a mild phenotype in *Caenorhabditis elegans* (*Staunton et al., 2001*). However, mutant mice do show increased synaptic recovery after intense stimulation, which is rescued by reintroducing full-length (FL) RPH3A but not a mutant that lacks the C2B domain. This suggests that RPH3A negatively regulates SV recycling after depletion of the releasable vesicle pool, and that this function depends on its interaction with SNAP25 (*Deák et al., 2006*). In *C. elegans*, the absence of RBF-1, the homolog of RPH3A, exacerbates the phenotypes of other SNARE mutants, as observed in the *rbf-1/ric-4* (RPH3A/SNAP25) double mutant, suggesting that RPH3A contributes to SNARE protein function (*Staunton et al., 2001*).

The role of RPH3A in the context of DCV exocytosis within mammalian neurons remains unclear. Notably, a recent genetic screen in *C. elegans* revealed that *rbf-1* depletion increases the release of fluorescently tagged neuropeptides without affecting the frequency of spontaneous mini-EPSCs. This suggest a potential negative regulatory role for RPH3A in neuropeptide release (*Laurent et al., 2018*), opposite to RIM1 (*Persoon et al., 2019*). However, it remains uncertain whether RPH3A functions as a negative regulator in mammalian neurons, and whether, similar to synaptic transmission, the interaction with SNAP25 is important.

We set out to determine the role of RPH3A in DCV exocytosis in hippocampal mouse neurons, and to investigate the relevance of RAB3A and SNAP25 binding. We found that RPH3A did not travel with DCVs, but remained stationary at synapses. Using RPH3A null mutant mice (*Schlüter et al., 1999*) we show that the absence of RPH3A indeed increased DCV exocytosis. Furthermore, RPH3A null neurons exhibit longer neurites and an increased number of DCVs. This effect was partly reduced when all regulated secretion was eliminated by tetanus neurotoxin (TeNT, *Hoogstraaten et al., 2020*; *Shimojo et al., 2015*), indicating that the increased neurite length partially depends on regulated secretion. Finally, expressing a mutant RPH3A unable to bind RAB3A/RAB27A restored DCV exocytosis, but not when expressing a mutant RPH3A unable to bind SNAP25. This suggests that limiting DCV exocytosis does not depend on the interaction with RAB3A, but at least in part on the interaction with SNAP25.

## Results

### RPH3A is enriched in presynaptic structures

RPH3A is reported to localize to both pre- and postsynaptic sites (*Stanic et al., 2015*), and on DCVs through its RAB-binding domain in PC12 cells (*Fukuda et al., 2004*). We first determined whether RPH3A is expressed in excitatory or inhibitory mouse neurons. To test this, we immunostained for

RPH3A in hippocampal VGLUT+ wildtype (WT) neurons and striatal VGAT+ WT neurons at 14 days in vitro (DIV14, *Figure 1—figure supplement 1A*). We found no difference in RPH3A expression between hippocampal and striatal neurons (*Figure 1—figure supplement 1B*). To localize RPH3A with higher spatial precision and confirm localization on DCVs, we performed stimulated emission depletion (STED) microscopy in hippocampal mouse neurons at DIV14. RPH3A staining was predominantly punctate and overlapped with the presynaptic marker Synapsin1 (Syn1, *Figure 1A and B*) and, to a lesser extent, with the postsynaptic marker Homer (*Figure 1A and C*). RPH3A partly colocalized with DCV marker chromogranin B (ChgB, *Figure 1A and D*). Pearson's correlation coefficients were similar for RPH3A with either Syn1, Homer, or ChgB (*Figure 1E*), however, Manders' overlap coefficient analysis (*Manders et al., 1992*) revealed that RPH3A colocalized significantly more with Syn1 than Homer or ChgB (*Figure 1F*). In addition, most RPH3A puncta contained Syn1, but not all Syn1 puncta contained RPH3A (*Figure 1G*). These data suggest that RPH3A is a synaptic protein with predominantly presynaptic accumulation.

We next determined which known interactions are important for RPH3A's synaptic localization by expressing mutant RPH3A constructs fused to mCherry, lacking specific interactions or a phosphorylation site, in hippocampal WT neurons at DIV1-2 (*Figure 1—figure supplement 1C*). FL RPH3A-mCherry and all mutant constructs were expressed to a similar level in neurites (*Figure 1—figure supplement 1G and H*). We performed STED microscopy at DIV14 and quantified the colocalization of mCherry with Syn1 and Homer (*Figure 1—figure supplement 1D and E*). Similar to endogenous RPH3A, FL WT RPH3A-mCherry showed a punctate distribution (*Figure 1—figure supplement 1D*) that colocalized strongly with Syn1, and to a lesser extent with Homer (*Figure 1—figure supplement 1F*). Three mutant versions of RPH3A: A C-terminal truncation (trunc. RPH3A), a $Ca^{2+}$-binding deficient mutant ($\Delta Ca^{2+}$-binding), and a mutant that does not bind SNAP25 ($\Delta$SNAP25), all retained a similar punctate presynaptic distribution (*Figure 1—figure supplement 1D and E*). However, mutant RPH3A unable to bind RAB3A/RAB27A ($\Delta$RAB3A/RAB27A, *Fukuda et al., 2004*) and a mutant that lacked a CaMKII-dependent phosphorylation site ($\Delta$CAMKII-phos. site) showed no, or less, punctate distribution (*Figure 1—figure supplement 1D and E*). These data suggest that the interaction with RAB3A is required for RPH3A's synaptic localization. CaMKII-dependent phosphorylation may also be involved in the synaptic localization of RPH3A.

## RPH3A does not travel with DCVs

Recent evidence has demonstrated that RAB3A is transported together with DCVs (*Persoon et al., 2019*). To test if RPH3A travels with DCVs on both axons and dendrites, we co-expressed neuropeptide Y (NPY)-mCherry and FL WT EGFP-RPH3A (*Tsuboi and Fukuda, 2005*) in DIV14 RPH3A knockout (KO) neurons. In addition, we tested co-transport of NPY fused to pH-sensitive EGFP (NPY-pHluorin), with two RPH3A mutants: a truncated RPH3A lacking both C2 domains but retaining the RAB3A and RAB27A-binding sites, and $\Delta$RAB3A/RAB27A mutant (*Figure 2A*, *Fukuda et al., 2004*). Live-cell imaging was performed before and after photobleaching a fixed-size area in neurites, without distinguishing between axons and dendrites, to enhance visualization of moving vesicles entering the bleached area (*Figure 2B–D*). The number of moving versus immobile puncta was determined for each construct prior to photobleaching. Both FL and truncated RPH3A exhibited low mobility (7–14% moving), although they occasionally showed small movements over short distances (less than 10 µm within 4 min). In contrast, 47% of NPY puncta were mobile (*Figure 2E*), as shown before (*Hoogstraaten et al., 2020*; *Persoon et al., 2019*). Only a small fraction of RPH3A (14%) and truncated RPH3A (7%) puncta traveled at velocities characteristic for DCV transport (*Bittins et al., 2010*; *de Wit et al., 2006*; *Kwinter et al., 2009*). These results indicate that the majority of RPH3A organizes in immobile puncta that colocalize with synaptic markers (see above), but do not travel through the axons or dendrites like DCVs do.

We next determined the fraction of RPH3A that traveled together with NPY-labeled vesicles. Few moving NPY vesicles contained RPH3A or truncated RPH3A (NPY:RPH3A=6%, NPY:trunc. RPH3A=1%). Conversely, of the already small fraction of moving FL and truncated RPH3A, only a few traveled together with NPY (RPH3A:NPY = 18%, truncated RPH3A:NPY = 8%; *Figure 2F*). Truncated RPH3A showed almost no fluorescence recovery after photobleaching compared to NPY (*Figure 2G*), however $\Delta$RAB3A/RAB27A RPH3A showed a faster recovery compared to FL and truncated RPH3A

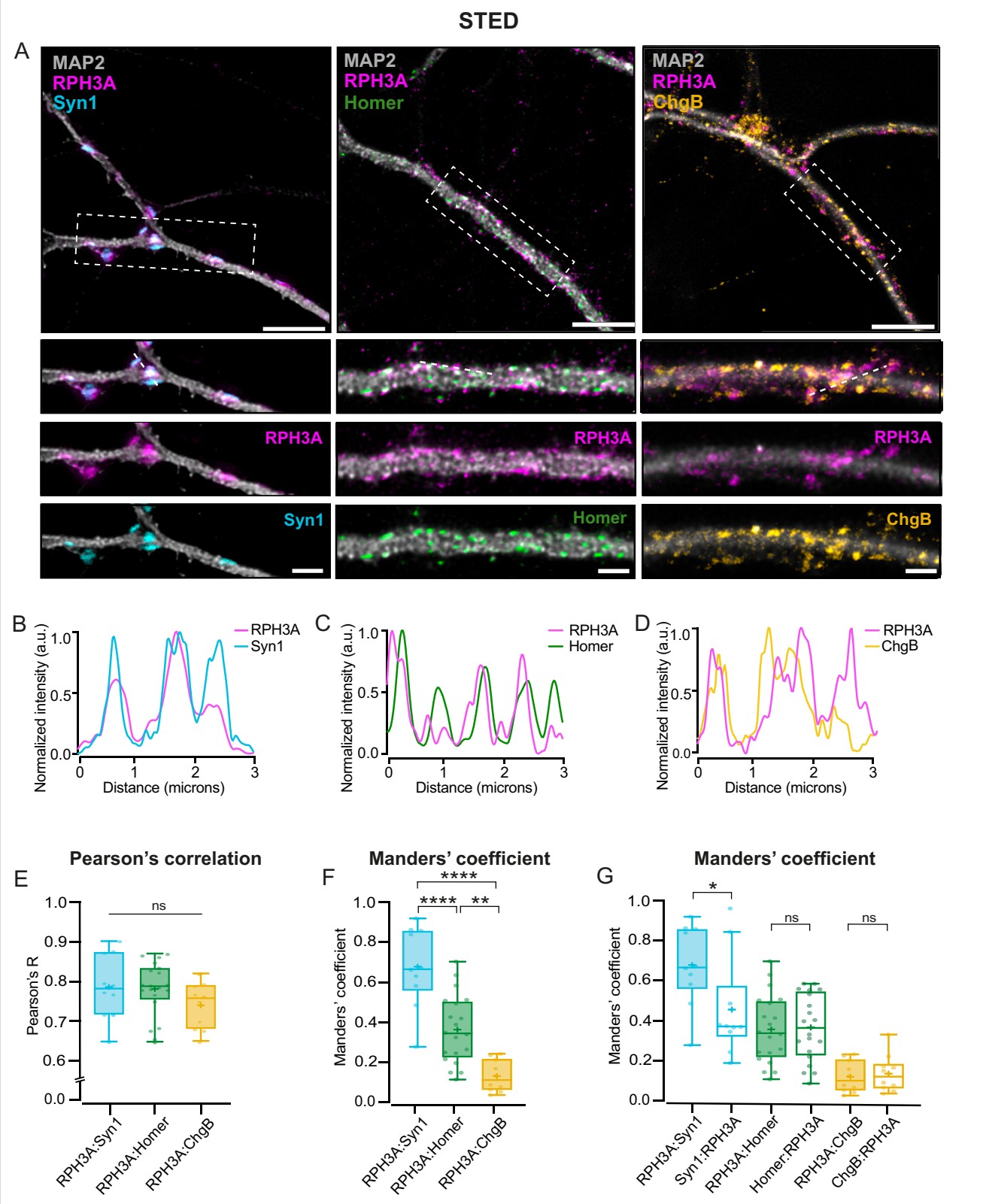

**Figure 1.** RPH3A localizes to the presynapses. (**A**) Representative images of wildtype (WT) hippocampal neurons (top) with zooms (bottom) co-stained for RPH3A (magenta) and Syn1 (cyan), Homer (green), or ChgB (yellow). Scale bar, 5 µm (top) and 2 µm (bottom). (**B–D**) Line plots show normalized fluorescent intensity across the dotted line in example zooms. Intensities are normalized from min to max. (**E**) Pearson's correlation coefficient and (**F**) Manders' overlap coefficient comparing the colocalization of RPH3A with Syn1, Homer, and ChgB. Each dot represents a field of view. N numbers of

*Figure 1 continued on next page*

*Figure 1 continued*

individual experiments: Syn1: 2 (10); Homer: 2 (20); ChgB: 2 (10). (**G**) Manders' overlap coefficient comparing the colocalization of RPH3A in either Syn1, Homer, or ChgB puncta, and vice versa. RPH3A:Syn1, RPH3A:Homer, and RPH3A:ChgB show the same dataset as in F. Boxplots represent the median (line), mean (+), and Tukey range (whiskers). Kruskal-Wallis H test with Dunn's correction: *p<0.05, **p<0.01, ****p<0.0001. ns = non-significant, p>0.05.

The online version of this article includes the following figure supplement(s) for figure 1:

**Figure supplement 1.** RPH3A localization depends on RAB3A/RAB27A-binding.

(*Figure 2H*). Overall, these findings suggest that RPH3A does not travel with DCVs and that the stationary organization of RPH3A relies on RAB3A/RAB27A interactions.

## RPH3A deficiency increases DCV exocytosis

Since RPH3A appeared to remain mostly stationary at the presynapse (*Figures 1 and 2*), we examined the role of RPH3A in neuropeptide release in hippocampal neurons. We recorded DCV fusion events in single hippocampal mouse neurons from RPH3A KO and WT littermates at DIV14–16. We confirmed the loss of RPH3A expression in RPH3A KO neurons with immunocytochemistry (*Figure 3A and B*). NPY-pHluorin, a validated DCV fusion reporter (*Arora et al., 2017*; *Farina et al., 2015*; *Persoon et al., 2018*; *van de Bospoort et al., 2012*), was used to quantify single fusion events (*Figure 3C*). To elicit DCV fusion, neurons were stimulated twice with 8 bursts of 50 action potentials (APs) at 50 Hz, separated by 30 s (*Figure 3D*) or once with 16 bursts of 50 APs at 50 Hz. The acidity of DCVs quenches NPY-pHluorin, but upon fusion with the plasma membrane, the DCV deacidifies resulting in increased fluorescent NPY-pHluorin intensity (*Figure 3C*). After stimulation, neurons were briefly perfused with $NH_4^+$ to de-quench all NPY-pHluorin labeled DCVs (*Figure 3C and D*) to determine the number of remaining DCVs per cell. We have previously demonstrated that the fluorescent intensity of NPY-pHluorin in confocal imaging directly correlates with the number of fluorescent puncta for endogenous DCV markers using super-resolution imaging (*Persoon et al., 2018*).

RPH3A KO neurons showed a threefold increase in the total number of fusion events compared to WT (*Figure 3F and G*, *Figure 3—figure supplement 1A and B*). To test if re-expressing RPH3A in KO neurons could restore DCV fusion to WT levels, we infected KO neurons with an FL RPH3A construct at DIV0 (*Figure 3E*). This construct localized to synapses (*Figure 1—figure supplement 1D*), similar to endogenous RPH3A expression (*Figure 1A–C*). Re-expression of RPH3A in KO neurons restored the number of fusion events to WT levels (*Figure 3F and G*). The released fraction, i.e., the number of fusion events divided by the remaining DCV pool, did not differ between WT and KO neurons (*Figure 3H*), and a trend toward a larger remaining DCV pool in these KO neurons was observed (*Figure 3—figure supplement 1C*).

We have recently shown that different stimulation protocols influence certain fusion dynamics like event duration, but not the total number of fusion events (*Baginska et al., 2023*). To test this and the robustness of our findings, we used a prolonged stimulation protocol (16 bursts of 50 APs at 50 Hz). The event duration was similar in WT and KO neurons, for both stimulation paradigms (*Figure 3—figure supplement 1D–F*). During prolonged stimulation, we observed an increase in DCV fusion events in KO neurons compared to WT (*Figure 3I and J*), similar to the 2×8 stimulation protocol. In addition, the released fraction was significantly increased in KO neurons compared to WT (*Figure 3K*). We did not observe a difference in remaining pool size between KO and WT in these neurons (*Figure 3—figure supplement 1G*). Overexpression of FL RPH3A in KO neurons again restored the number and released fraction of fusion events to WT levels (*Figure 3I–K*). We did not observe an effect on spontaneous DCV fusion in RPH3A KO neurons (*Figure 3—figure supplement 1H*). Together, these findings indicate that RPH3A is an inhibitor of DCV exocytosis.

## The SNAP25, but not the RAB3A interaction domain of RPH3A contributes to limiting DCV exocytosis

To investigate whether the interactions with RAB3A or SNAP25 are relevant to limit DCV exocytosis, we overexpressed FL WT RPH3A, ΔRAB3A/RAB27A mutant RPH3A (*Fukuda et al., 2004*), and ΔSNAP25 mutant RPH3A (*Ferrer-Orta et al., 2017*) in RPH3A KO neurons (*Figure 4A*). Expression of ΔRAB3A/RAB27A restored DCV fusion and the released fraction to WT levels, similar to FL RPH3A (*Figure 4B, D, and E*). However, expression of ΔSNAP25 did not fully rescue the number of

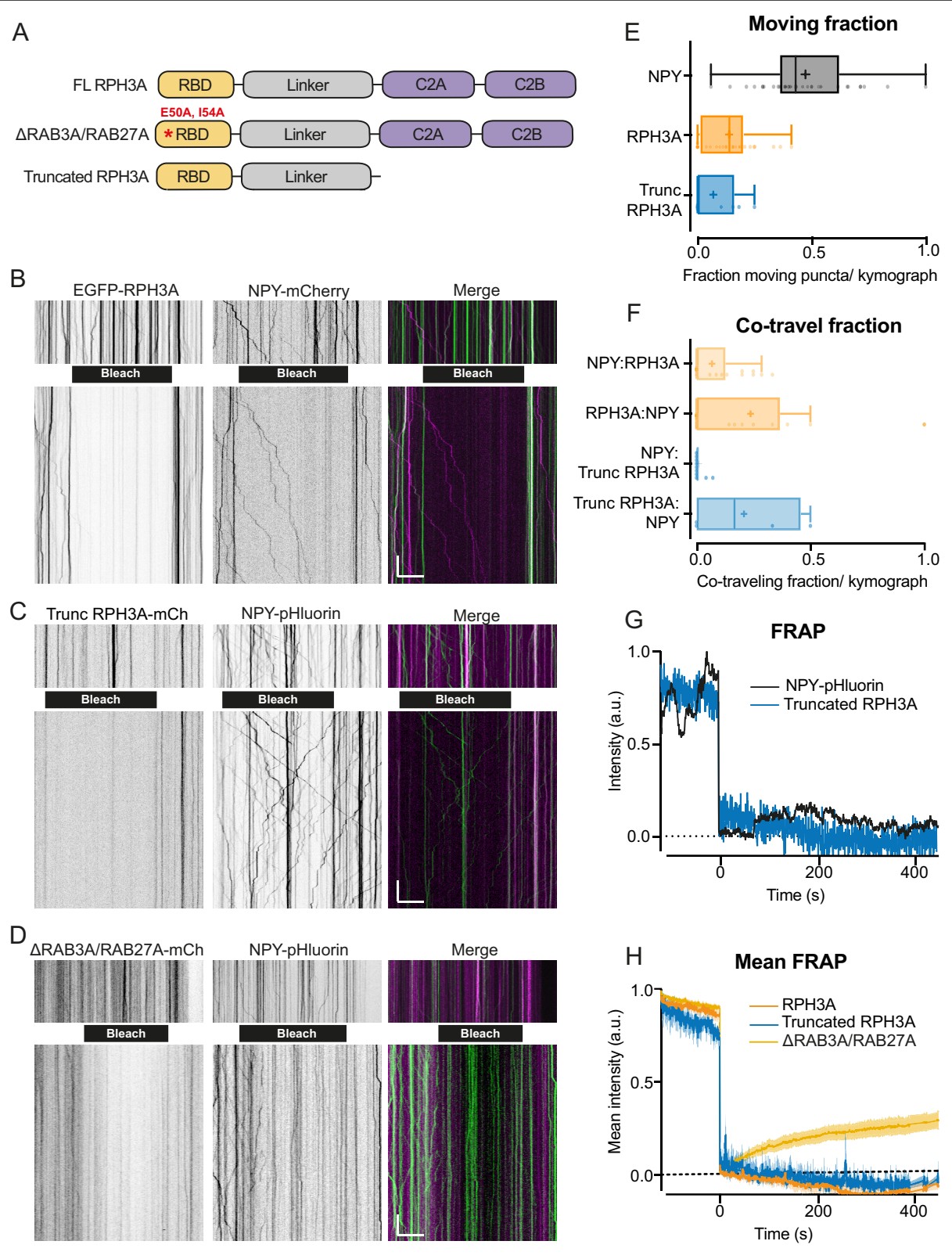

**Figure 2.** RPH3A does not travel with dense core vesicles (DCVs). (**A**) Domain structures of full-length (FL) RPH3A and mutant RPH3A constructs lacking specific interactions: ΔRAB3A/RAB27A mutant RPH3A and truncated RPH3A that lacked its calcium and SNAP25-binding C2A and C2B domain. (**B**) Kymographs of EGFP-RPH3A and neuropeptide Y (NPY)-mCherry, (**C**) mCherry-trunc. RPH3A and NPY-pHluorin, and (**D**) mCherry-ΔRAB3A/RAB27A mutant RPH3A and NPY-pHluorin before (upper) and after (lower) photobleaching (black bar). NPY-pHluorin showed more resistance to bleaching. This

*Figure 2 continued on next page*

*Figure 2 continued*

posed no issue as bleaching was merely applied to enhance the visualization of vesicles entering the bleached area and facilitate analysis. Merged images show mCherry (pseudo-colored magenta) and EGFP/pHluorin (green). Scale bar, 20 µm (x-axis) and 20 s (y-axis). (**E**) Moving fraction of NPY, FL RPH3A, and truncated RPH3A puncta per kymograph. N numbers of individual experiments: NPY: 1 (28); RPH3A: 1 (38); trunc. RPH3A: 1 (10). Dots represent a kymograph. (**F**) Fraction of co-travel of NPY puncta with either FL or truncated RPH3A puncta, and co-travel of FL or truncated RPH3A with NPY puncta. N numbers of individual experiments: NPY:RPH3A: 1 (28); RPH3A:NPY: 1 (21); NPY:trunc. RPH3A: 1 (10); trunc. RPH3A:NPY: 1 (4). (**G**) Fluorescent recovery of the traces shown in C after photobleaching truncated RPH3A or NPY-pHluorin, normalized from min to max. (**H**) Mean fluorescent recovery traces from multiple kymographs after photobleaching FL, truncated, or ΔRAB3A/RAB27A RPH3A. Lines±shading represents mean ± SEM. Boxplots represent median (line), mean (+), and Tukey range (whiskers).

fusion events or the released fraction (*Figure 4C, G, and H*). We did not observe any difference in the number of remaining DCVs upon ΔRAB3A/RAB27A or ΔSNAP25 expression (*Figure 4F and I*). Taken together, these results suggest that RAB3A/RAB27A binding is not essential for the limiting effect of RPH3A on DCV exocytosis, but that the interaction with SNAP25 appears to contribute to this effect.

## RPH3A deficiency leads to increased neurite length and DCV numbers

Since we observed a trend toward a bigger DCV pool in KO neurons (*Figure 3—figure supplement 1C*), and the total number of DCVs per neuron correlates with dendrite length (*Persoon et al., 2018*), we examined the neuronal morphology of RPH3A KO neurons. Single hippocampal neurons (DIV14) were immunostained for MAP2 to quantify the total dendritic length, and for endogenous DCV cargo ChgB to determine the number of DCVs (*Figure 5A*). We have previously shown that this correlates well with the number of dSTORM ChgB puncta (*Persoon et al., 2018*). Indeed, neurons lacking RPH3A had longer dendrites that harbored more DCVs than WT neurons (*Figure 5B and C*) and KO neurons contained more DCVs per µm (*Figure 5D*). The number of DCVs correlated with the total dendrite length in both genotypes (*Figure 5F*), as shown previously for WT neurons (*Persoon et al., 2018*). Moreover, the intensity of endogenous ChgB was decreased in RPH3A KO neurons (*Figure 5E*), suggesting affected vesicle loading or reduced clustering. However, the peak intensity of fusion events during live recording was unchanged (*Figure 3—figure supplement 1I*), indicating that the decrease in DCV cargo intensity is potentially due to reduced clustering or accumulation. Taken together, these results suggest that lack of RPH3A leads to longer dendrites with a concomitant increase in the number of DCVs. RPH3A depletion does not seem to affect the neuropeptide content in DCVs, instead it might reduce the clustering of DCVs.

## Increased neurite length upon RPH3A deficiency partly depends on regulated secretion

RPH3A deletion resulted in increased DCV exocytosis (*Figure 3*) and dendrite length (*Figure 5B*). We reasoned that increased release of neuropeptide and neurotrophic factors throughout development could contribute to the longer neurites observed. To test this, we inhibited both SV and DCV exocytosis by cleaving VAMP1, VAMP2, and VAMP3 using TeNT (*Hoogstraaten et al., 2020*; *Humeau et al., 2000*), followed by immunostainings for MAP2 and Tau to assess dendritic and axonal length, respectively. Neurons infected with TeNT at DIV1 lacked VAMP2 staining at DIV14, confirming successful cleavage (*Figure 6A*). TeNT expression in WT neurons had no effect on dendritic (*Figure 6B*) or axonal length (*Figure 6C*), as shown before (*Harms and Craig, 2005*). TeNT expression in KO neurons restored neurite length to WT levels (*Figure 6B and C*). When comparing KO neurons with and without TeNT, KO neurons with TeNT show a trend toward decreased neurite length, similar to WT (*Figure 6C and B*). Re-expression of RPH3A in KO neurons without TeNT restored dendritic and axonal length to WT levels (*Figure 6B and C*). To identify the downstream pathway, we overexpressed mutant RPH3A constructs, lacking specific interactions (*Figure 1—figure supplement 1C*). No significant differences in dendrite or axon length were observed for any of the mutants compared to WT (*Figure 6—figure supplement 1*). These results indicate that regulated secretion is not required for neurite outgrowth, but that the increased neurite length upon RPH3A depletion depends, at least in part, on regulated secretion.

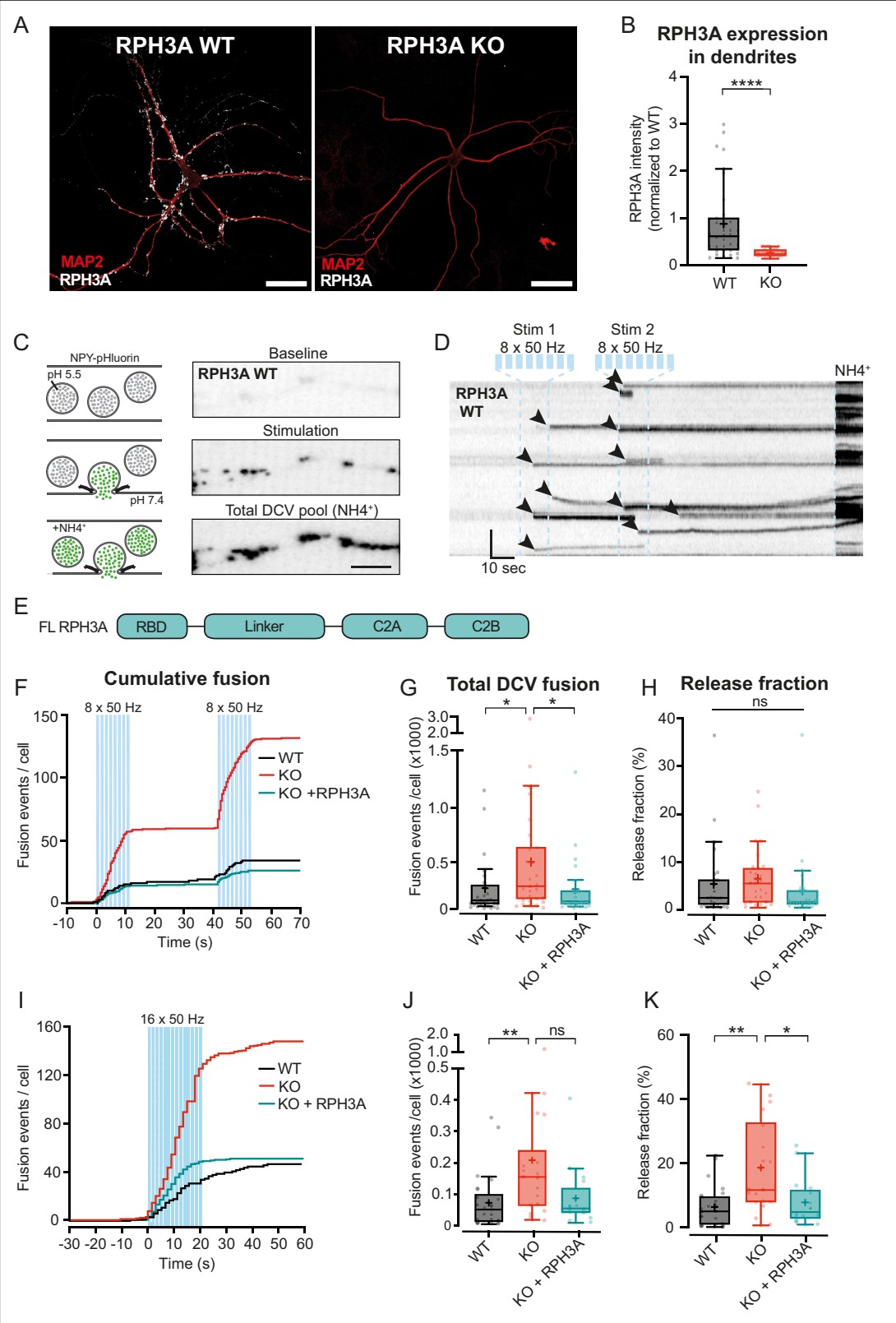

**Figure 3.** RPH3A deficiency increases dense core vesicle (DCV) exocytosis. (**A**) Typical example of RPH3A wildtype (WT) and knockout (KO) neurons immunostained for MAP2 (red) and RPH3A (white). Scale bar, 50 μm. (**B**) RPH3A expression in dendrites of WT and KO neurons normalized to WT per independent experiment. N numbers of individual experiments and single neuron observations in brackets: WT: 4 (34); KO: 4 (33). (**C**) Schematic representation (left) and imaging example of a WT neurite stretch (right) infected with neuropeptide Y (NPY)-pHluorin as optical DCV fusion reporter.

*Figure 3 continued on next page*

*Figure 3 continued*

NPY-pHluorin is quenched in the acidic DCV lumen before fusion (baseline) but dequenches upon fusion (stimulation). NH4⁺ perfusion dequenches all NPY-pHluorin labeled DCVs (remaining DCV pool). Scale bar, 5 μm. (**D**) Kymograph of a WT neurite stretch with the stimulation paradigm used to elicit DCV fusion (two bursts of 8×50 action potential (AP) trains at 50 Hz interspaced by 0.5 s between each train and 30 s between each burst, blue bars) and NH4 perfusion (NH4⁺) used to dequench all NPY-pHluorin labeled vesicles. Arrowheads indicate fusion events. Scale bar, 10 s. (**E**) Domain structure of full-length (FL) RPH3A construct. (**F**) Cumulative median histogram of fusion events over time in WT (black), RPH3A KO (red), and KO neurons infected with FL RPH3A (cyan). Blue bars indicate the stimulation paradigm (two bursts of 8×50 AP bursts at 50 Hz). (**G**) Total number of DCV fusion events per condition (two bursts of 8×50 AP bursts at 50 Hz). (**H**) Released fraction defined as the number of fusion events normalized to the remaining pool of DCVs. N numbers of individual experiments and single neuron observations in brackets: WT: 5 (51); KO: 5 (43); KO+RPH3A: 5 (39). (**I**) Cumulative median histogram of events over time in WT (black), RPH3A KO (red), and KO neurons infected with FL RPH3A (cyan). Blue bars indicate the stimulation paradigm (16×50 AP bursts at 50 Hz). (**J**) Total number of DCV fusion events per condition (16×50 AP bursts at 50 Hz). (**K**) Release fraction per cell. N numbers of individual experiments and single neuron observations in brackets: WT: 4 (25); KO: 4 (18); KO+RPH3A: 4 (16). Boxplots represent the median (line), mean (+), and Tukey range (whiskers). Each dot represents an individual neuron. Line graphs represent the median. Mann-Whitney U test and or Kruskal-Wallis H test with Dunn's correction: *p<0.05, **p<0.01, ****p<0.0001. ns = non-significant, p>0.05.

The online version of this article includes the following figure supplement(s) for figure 3:

**Figure supplement 1.** RPH3A depletion increases dense core vesicle (DCV) exocytosis, but does not affect remaining DCV pool size or content.

## Discussion

In this study, we addressed the role of RPH3A in DCV exocytosis in hippocampal neurons. RPH3A predominantly localized to the presynapse (*Figure 1*) and did not travel with DCVs (*Figure 2*). RPH3A null mutant neurons showed an increase in DCV exocytosis (*Figure 3*). Expression of a RAB3A/RAB27A-binding deficient, but not a SNAP25-binding deficient RPH3A, in RPH3A KO neurons, restored DCV exocytosis to WT levels (*Figure 4*). Finally, RPH3A null neurons had longer neurites that contained more DCVs (*Figure 5*). The increase in neurite length may partially depend on regulated secretion, as TeNT expression showed a strong trend toward reduced neurite length to WT levels (*Figure 6*). Taken together, we conclude that RPH3A limits DCV exocytosis, partly through its interaction with SNAP25.

### Presynaptic enrichment and accumulation of RPH3A

We found that RPH3A is enriched at the presynapse. This is in line with previous studies showing synaptic enrichment of RPH3A (*Li et al., 1994*; *Mizoguchi et al., 1994*; *Stanic et al., 2015*; *Stanic et al., 2017*). RPH3A showed mostly a punctate distribution, except in the presynapse, where it tends to accumulate (*Figure 1A*). Previous studies showed that inactivation of *synapsin1* and *2* genes decreased RPH3A levels but increased phosphorylation of the remaining RPH3A (*Lonart and Simsek-Duran, 2006*). Hence, synapsins may, directly or indirectly, contribute to presynaptic RPH3A accumulation, for instance as part of the phase separation of the vesicle cluster. In addition, phosphorylation reduces RPH3A's affinity for membranes (*Foletti et al., 2001*) and therefore preferential presynaptic dephosphorylation may also help to explain the presynaptic RPH3A accumulation.

### RPH3A does not travel with DCVs in hippocampal neurons

We demonstrated that FL and truncated RPH3A are highly stationary at synapses. However, a RPH3A mutant unable to bind RAB3A/RAB27A was more mobile (faster recovery after photobleaching, *Figure 2*), and lost its punctate distribution. This indicates that the synaptic enrichment of RPH3A depends, at least in part, on RAB3A/RAB27A interactions. This is in line with previous findings showing that the expression levels and localization of RPH3A in mammalian neurons are dependent on RAB3 (*Schlüter et al., 1999*). These data do not exclude that RPH3A interacts with immobile DCVs at synapses, potentially in a docked or primed state at the release sites, consistent with our conclusion that RPH3A serves as a negative regulator of DCV exocytosis. Based on these findings, we propose that RAB3A recruits RPH3A to DCV release sites, where it interacts with (immobile) DCVs and potentially competes with essential DCV proteins, such as synaptobrevin/VAMP2 and synaptotagmin-1/synaptotagmin-7 (*Hoogstraaten et al., 2020*; *van Westen et al., 2021*), for SNAP25 binding (*Figure 7*).

### RPH3A limits DCV exocytosis in hippocampal neurons

DCV exocytosis was increased by threefold upon RPH3A depletion and this effect was rescued by overexpressing FL RPH3A (*Figure 3F and I*). The increase in released fraction upon RPH3A depletion

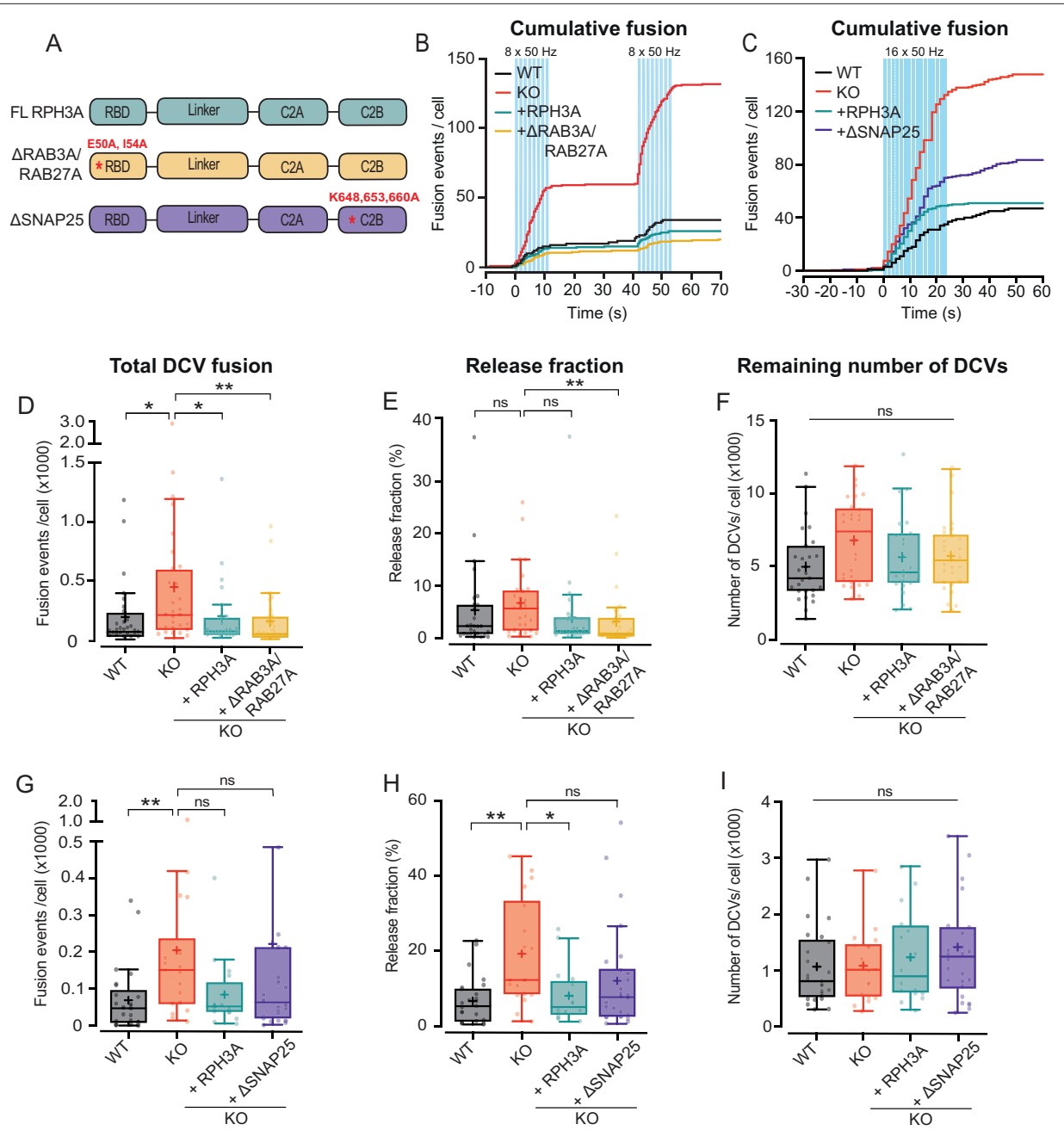

**Figure 4.** RPH3A interaction with SNAP25, but not RAB3A, partly contributes to limiting dense core vesicle (DCV) exocytosis. This figure shows the same dataset for wildtype (WT), knockout (KO), and full-length (FL) RPH3A as in *Figure 3*. (**A**) Domain structures of FL RPH3A (cyan), ΔRAB3A/RAB27A (yellow), and ΔSNAP25 mutant RPH3A (purple) with the corresponding mutant sites in red. (**B**) Cumulative median histogram of events over time in WT (black), RPH3A KO (red), and KO neurons infected with FL RPH3A (cyan) or ΔRAB3A/RAB27A mutant RPH3A (yellow). Blue bars indicate the stimulation paradigm (two bursts of 8×50 action potential [AP] bursts at 50 Hz). (**C**) Cumulative median histogram of events over time in WT (black), KO (red), and KO neurons infected with FL RPH3A (cyan) or ΔSNAP25 mutant RPH3A (purple). Blue bars indicate the stimulation paradigm (16×50 AP bursts at 50 Hz). (**D**) Total DCV fusion events in WT (black), KO (red), and KO neurons expressing RPH3A (cyan) or ΔRAB3A/RAB27A (yellow). (**E**) Released fraction of the number of fusion events normalized to the remaining DCV pool per cell. Expression of ΔRAB3A/RAB27A in KO neurons significantly decreased the number of fusion events and released fraction to WT levels. ΔRAB3A/RAB27A did not differ from WT or FL RPH3A. (**F**) Remaining neuropeptide Y (NPY)-pHluorin labeled DCV pool estimates derived from NH4+ perfusion after stimulation. N numbers of individual experiments and single neuron observations in brackets: WT: 4 (27); KO: 4 (28); KO+RPH3A: 4 (24); KO+ΔRAB3A/RAB27A: 4 (28). (**G**) Total number of DCV fusion events in WT (black), KO (red), and KO neurons expressing RPH3A (cyan) or ΔSNAP25 (purple). (**H**) Release fraction per cell. ΔSNAP25 expression in KO neurons was unable to fully rescue the number of fusion events and released fraction to WT levels. ΔSNAP25 did not significantly differ from WT or FL RPH3A. (**I**) Remaining

*Figure 4 continued on next page*

*Figure 4 continued*

NPY-pHluorin labeled DCV pool per cell. N numbers of individual experiments and single neuron observations in brackets: WT: 4 (25); KO: 4 (18); KO+RPH3A: 4 (16); KO+ΔSNAP25: 4 (26). Line graphs represent the median. Boxplots show the mean (+), median (line), and Tukey range (whiskers). Each dot represents a single neuron. Kruskal-Wallis H test with Dunn's correction: *p<0.05, **p<0.01. ns = non-significant, p>0.05.

was most significant during intense, prolonged stimulation (16 bursts of 50 APs, *Figure 3K*). During more distributed stimulation, a similar trend was observed (*Figure 3F–H*). Together, these results suggest that RPH3A is a negative regulator of DCV exocytosis (*Figure 7A and B*). This is in line with *C. elegans* data, showing that the lack of the nematode homolog *rbf-1* increased DCV exocytosis, but had no effect on spontaneous release of SVs. This suggests that *rbf-1* limits DCV exocytosis only (*Laurent et al., 2018*). In addition, expression of FL RPH3A in PC12 cells reduced high KCl-dependent neuropeptide release (*Fukuda et al., 2004*). In this study, we demonstrated that RPH3A also negatively regulated neuropeptide release in mammalian hippocampal neurons. Our prior estimations in mouse hippocampal neurons indicated that merely 1–6% of the total DCV pool undergo exocytosis upon strong stimulation (*Persoon et al., 2018*), implying the existence of an inhibitory release mechanism. To our knowledge, RPH3A is the only negative regulator of DCV exocytosis in mammalian neurons identified so far, without substantial impact on SV exocytosis.

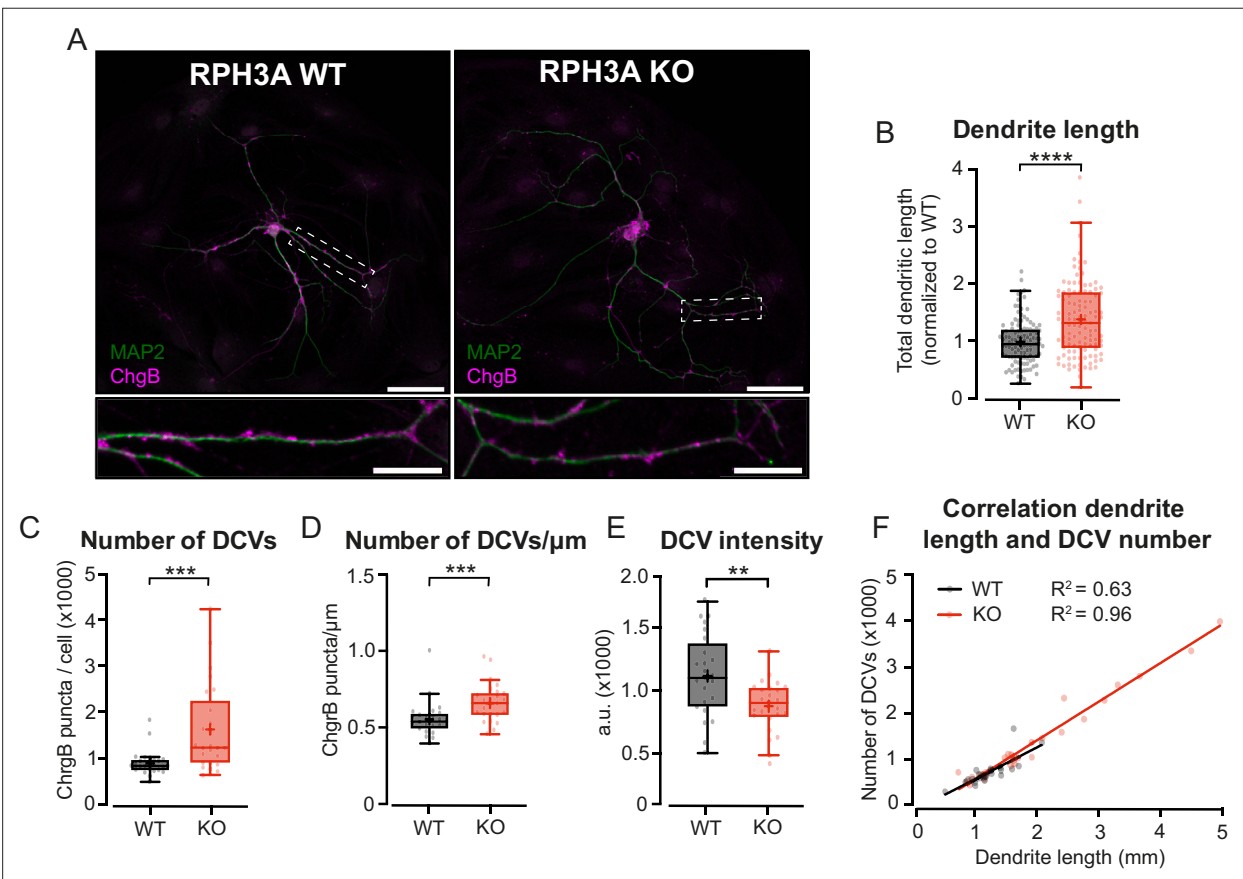

**Figure 5.** Increased neurite length and dense core vesicle (DCV) number upon RPH3A deficiency. (**A**) Typical example of a single wildtype (WT) and RPH3A knockout (KO) hippocampal neuron (top) with zooms (bottom) stained for MAP2 (green) and the DCV marker ChgB (magenta). Scale bars, 50 μm (top) and 20 μm (bottom). (**B**) Total dendritic length of single hippocampal RPH3A WT or KO neurons normalized to WT per independent experiment. N numbers of individual experiments and single neuron observations in brackets: WT: 14 (112); KO: 14 (113). (**C**) Total number of ChgB labeled DCVs per neuron for each group. N numbers per condition: WT: 3 (24); KO: 3 (25). (**D**) Total ChgB labeled DCVs per μm for each neuron per group. (**E**) Mean intensity of ChgB labeled DCVs per neuron for each group. (**F**) Correlation between ChgB labeled DCVs and dendritic length (mm). Linear regression goodness of fit ($r^2$) is given for each group. Boxplots represent median (line), mean (+), and Tukey range (whiskers). Each dot represents an individual neuron. Mann-Whitney U or unpaired t-test: **p<0.01, ***p<0.001, ****p<0.0001.

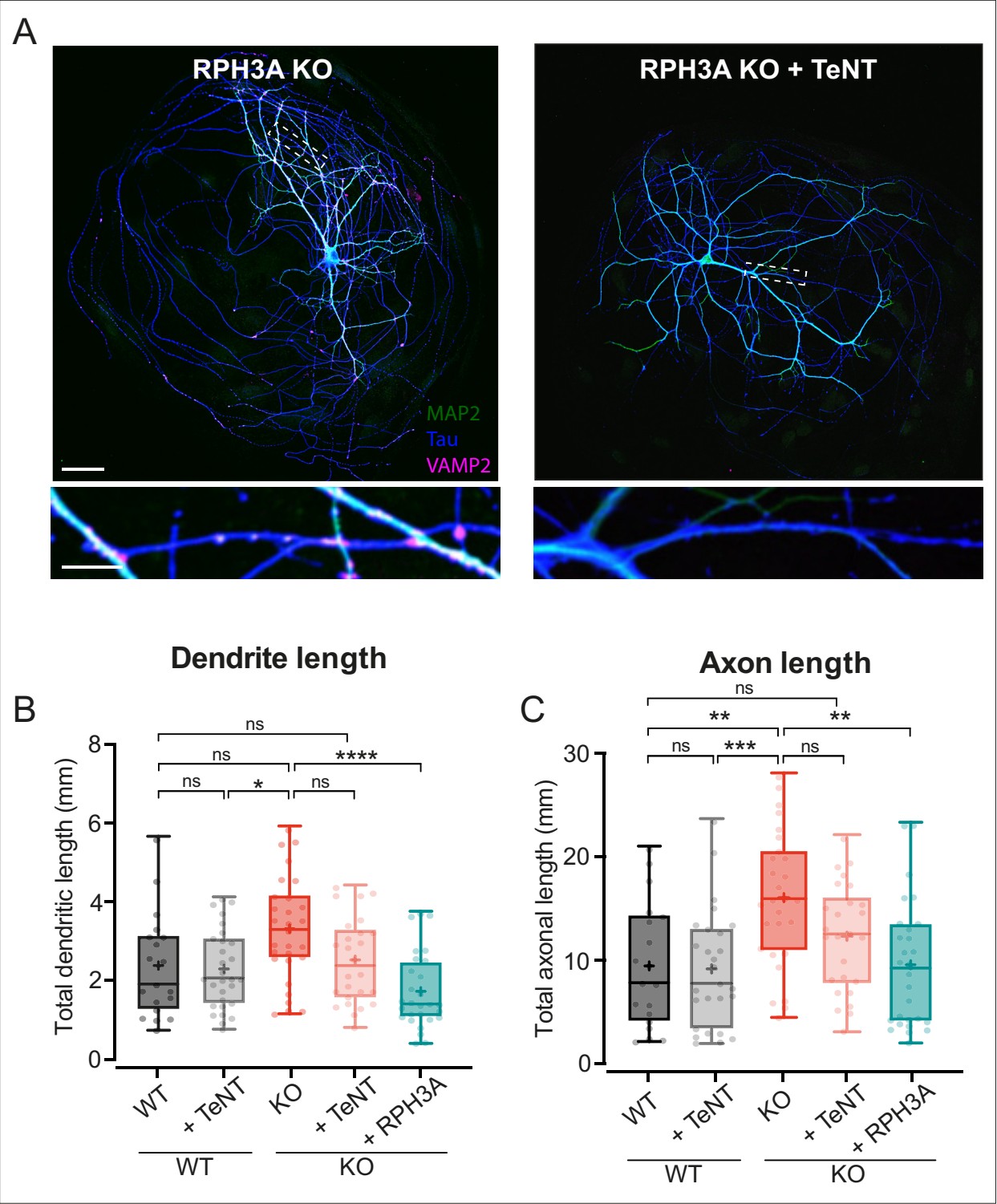

**Figure 6.** Increased neurite length upon RPH3A deficiency partly depends on regulated secretion. (**A**) Typical example of a single RPH3A knockout (KO) neurons either infected with tetanus neurotoxin (TeNT) or not, showing successful VAMP2 cleavage with zooms (bottom) stained for MAP2 (green), Tau (blue), and VAMP2 (magenta). Scale bars, 50 μm (top) and 20 μm (bottom). (**B**) Total dendritic and (**C**) axonal length (mm) of wildtype (WT) and KO neurons -/+TeNT, and KO neurons expressing RPH3A. N numbers of individual experiments and single neuron observations in brackets: WT: 3 (19); WT+TeNT: 3 (27); KO: 3 (28). KO+TeNT: 3 (26), KO+RPH3A: 3 (28). Boxplots represent median (line), mean (+), and Tukey range (whiskers). Each dot represents an individual neuron. Kruskal-Wallis H test with Dunn's correction: *p<0.05, **p<0.01, ***p<0.001, ****p<0.0001. ns = non-significant, p>0.05.

The online version of this article includes the following figure supplement(s) for figure 6:

*Figure 6 continued on next page*

*Figure 6 continued*

**Figure supplement 1.** Increased neurite length upon RPH3A depletion does not depend on RAB3A/RAB27A binding, calcium binding, or phosphorylation of RPH3A.

## The limiting effect of RPH3A on DCV exocytosis partially depends on SNAP25 binding

We recently discovered that RAB3 and its effector RIM1 are positive regulators of DCV exocytosis in mammalian hippocampal neurons (*Persoon et al., 2019*). In contrast, we demonstrated that RPH3A serves as a negative regulator of DCV exocytosis. Given that RPH3A is a downstream effector of RAB3, and that RAB3 is necessary for synaptic RPH3A enrichment, we expected that the interaction with RAB3 plays a role in limiting DCV exocytosis. However, enhanced DCV exocytosis was rescued upon expression of a RPH3A mutant that was unable to bind RAB3A/RAB27A, suggesting that the limiting effect of RPH3A on DCV exocytosis is independent of an interaction with RAB3A (*Figure 7C*). This indicates that even though RAB3A is important for the localization of RPH3A, RPH3A can still limit exocytosis when its punctate organization is lost, suggesting that the interaction with RAB3A is not rate limiting. One other potential mechanism by which RPH3A could directly limit exocytosis is by limiting SNAP25 availability. Previous research has demonstrated that RPH3A binding to SNAP25 negatively regulates SV recycling (*Deák et al., 2006*). We found that expressing mutant RPH3A that lacks SNAP25 binding in KO neurons does not fully restore DCV exocytosis to WT levels. This suggests that RPH3A limits DCV exocytosis by interacting with SNAP25 (*Figure 7D*). The partial selectivity for the DCV secretory pathway may be attributed to RPH3A functioning as a downstream effector of RAB3A. RAB3 is essential for DCV exocytosis (*Persoon et al., 2019*) but largely redundant for synaptic

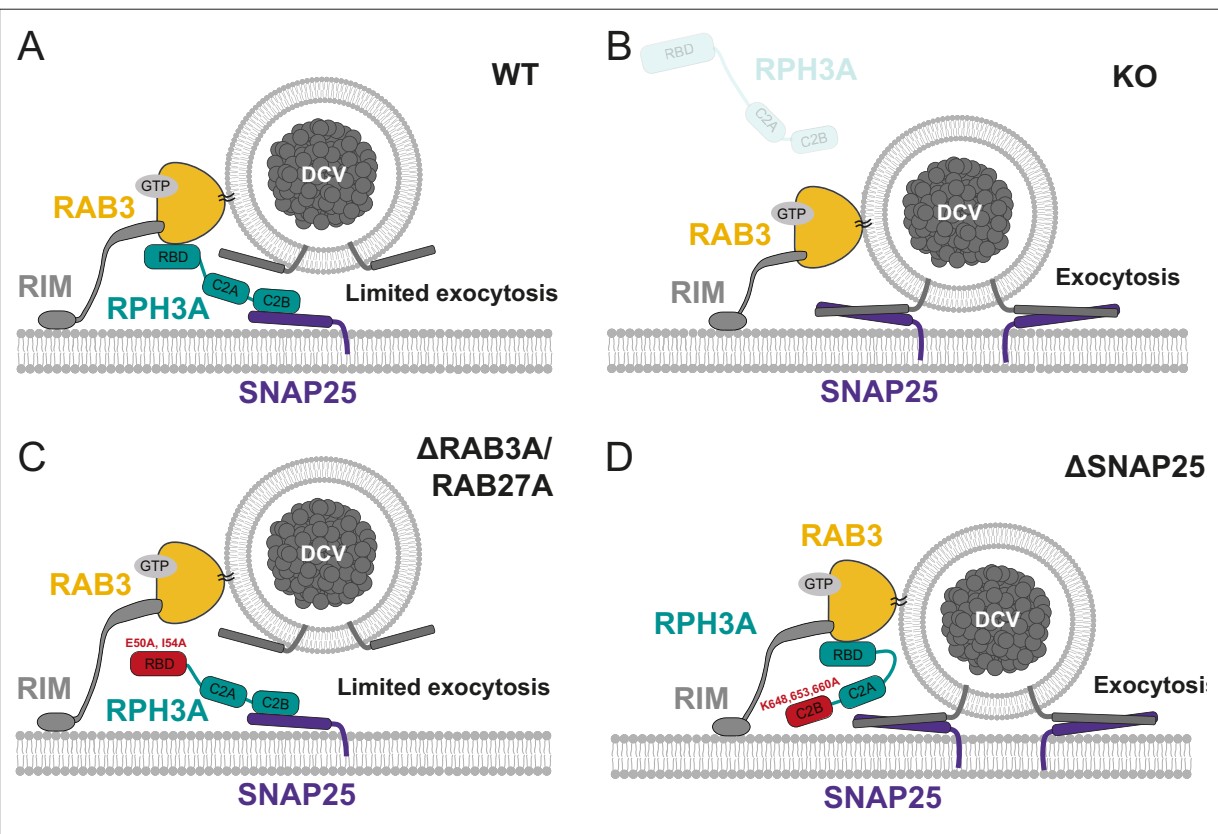

**Figure 7.** Role of RPH3A in dense core vesicle (DCV) exocytosis. (**A**) RPH3A binding to RAB3 through its RAB-binding domain (RBD) ensures confined presynaptic localization of RPH3A. RPH3A binding to SNAP25 through its C2B domain inhibits DCV exocytosis, potentially by inhibiting SNAP25 binding of essential DCV proteins synaptobrevin/VAMP2 and/or synaptotagmin (not depicted). (**B**) In the absence of RPH3A, DCV exocytosis is not limited. (**C**) Upon expression of a RPH3A mutant that is unable to bind RAB3A/RAB27A, DCV exocytosis is limited. (**D**) When RPH3A is unable to bind SNAP25, the SNARE assembly is not restricted and therefore DCV exocytosis is not limited, while RPH3A is still recruited to synapses/release sites via RAB3.

transmission (*Schlüter et al., 2006*; *Schlüter et al., 2004*). Based on our findings, we propose that RAB3A plays a role in recruiting RPH3A to synaptic exocytosis sites, where RPH3A might bind available SNAP25, potentially restricting the assembly of SNARE complexes and thereby inhibiting DCV exocytosis (*Figure 7*).

## Increased regulated secretion in RPH3A KO neurons might lead to longer neurites

RPH3A KO neurons have longer neurites that correlated with the number of DCVs as shown before (*Persoon et al., 2018*). In agreement with previous findings (*Ahnert-Hilger et al., 1996*; *Grosse et al., 1999*; *Osen-Sand et al., 1996*), we find that TeNT expression did not affect neurite length of WT neurons, but showed a trend toward shorter neurites in RPH3A KO neurons. Based on this, we speculate that the increased neurite length in RPH3A KO neurons might, at least partially, be driven by TeNT-dependent regulated secretion, in particular VAMP1, 2, or 3 mediated exocytosis. Given that neuropeptides and neurotrophic factors are key modulators of neuronal maturation and outgrowth (*Mu et al., 2010*), and that RPH3A depletion leads to increased DCV exocytosis, it stands to reason that we observed longer neurites in RPH3A KO neurons.

The partial rescue by TeNT suggests RPH3A-dependent mechanisms that could explain the increase in neuron size besides regulated secretion. RPH3A could control neurite length by regulating the actin cytoskeleton. RPH3A interacts with actin via binding to α-actinin and β-adducin (*Baldini et al., 2005*; *Coppola et al., 2001*; *Kato et al., 1996*). In vitro experiments showed that RPH3A together with α-actinin can bundle actin (*Kato et al., 1996*). Regulation of the actin cytoskeleton has extensively been linked to neurite outgrowth (*Meberg and Bamburg, 2000*) making it plausible that lack of RPH3A alters actin regulation, resulting in longer neurites.

# Materials and methods

## Key resources table

| Reagent type (species) or resource | Designation | Source or reference | Identifiers | Additional information |
|---|---|---|---|---|
| Genetic reagent (*Mus musculus*) | C57BL/6J | Charles River | Strain code 631 | |
| Genetic reagent (*Mus musculus*) | *Rph3a⁻/⁻* mice | *Schlüter et al., 1999* | – | See section 'Animals' |
| Antibody | Anti-chromogranin B (rabbit polyclonal) | SySy | 259103 | 1:500 |
| Antibody | Anti-RPH3A (mouse monoclonal) | Transduction Laboratories | – | 1:1000 |
| Antibody | Anti-MAP2 (chicken polyclonal) | Abcam | ab5392 | 1:500 |
| Antibody | Anti-Syn1 (rabbit polyclonal) | #P610; SySy | 106 103 | 1:1000; 1:500 |
| Antibody | Anti-VGLUT (rabbit polyclonal) | SySy | 135302 | 1:500 |
| Antibody | Anti-Homer1 (guinea pig polyclonal) | SySy | 160 004 | 1:500 |
| Antibody | Anti-Tau (xx, polyclonal) | SySy | 314 004 | 1:1000 |
| Antibody | Anti-VAMP2 (mouse monoclonal) | SySy | 104 211 | 1:1000 |
| Antibody | Anti-mCherry (mouse monoclonal) | Signalway Antibody | #T515 | 1:1000 |
| Recombinant DNA reagent | pSyn(pr)Rabphilin3a(mus)-IRES2NLSCherryLL3.7 | This paper | - | Generation of this reagent is described in Materials and methods section 'Lentiviral vectors and infections' |
| Recombinant DNA reagent | pSyn(pr)EGFP-Glyrnlinker-Rabphilin3A-lentiFGA2.0 | This paper | - | Generation of this reagent is described in Materials and methods section 'Lentiviral vectors and infections' |

*Continued on next page*

*Continued*

| Reagent type (species) or resource | Designation | Source or reference | Identifiers | Additional information |
|---|---|---|---|---|
| Recombinant DNA reagent | pSyn(pr)mCherry-Glyrnlinker-Rabphilin3A(E50A,I54A) lenti-FGA2.0 | This paper | - | Generation of this reagent is described in Materials and methods section 'Lentiviral vectors and infections' |
| Recombinant DNA reagent | pSyn(pr)mScarlet-Glyrnlinker-Rabphilin3A(K648,653,660A)lentiFGA2.0 | This paper | - | Generation of this reagent is described in Materials and methods section 'Lentiviral vectors and infections' |
| Recombinant DNA reagent | pSyn(pr)mCherry-Glyrnlinker-Rabphilin3A(1-375)lenti-FGA2.0 | This paper | - | Generation of this reagent is described in Materials and methods section 'Lentiviral vectors and infections' |
| Recombinant DNA reagent | pSyn(pr)mCherry-Glyrnlinker-Rabphilin3A(D568N,D574N) lentiFGA2.0 | This paper | - | Generation of this reagent is described in Materials and methods section 'Lentiviral vectors and infections' |
| Recombinant DNA reagent | pSyn(pr)mCherry-Glyrnlinker-Rabphilin3A(S271A)-lentiFGA2.0 | This paper | - | Generation of this reagent is described in Materials and methods section 'Lentiviral vectors and infections' |
| Recombinant DNA reagent | pSyn(pr)hNPYPHluorin-N1lenti | *van de Bospoort et al., 2012* | - | Generation of this reagent is described in Materials and methods section 'Lentiviral vectors and infections' |
| Recombinant DNA reagent | pSyn(pr)hNPYCherryLenti | *de Wit et al., 2009*; *Farina et al., 2015* | - | Generation of this reagent is described in Materials and methods section 'Lentiviral vectors and infections' |
| Recombinant DNA reagent | pSyn(pr)HA-Tetx(E234Q)IRES2CherryDEST-lenti-fga2.0 | *Emperador Melero et al., 2017* | - | Generation of this reagent is described in Materials and methods section 'Lentiviral vectors and infections' |
| Software, algorithm | MATLAB | MathWorks | - | |
| Software, algorithm | Prism | GraphPad | - | |
| Software, algorithm | ImageJ/Fiji | ImageJ | - | |
| Software, algorithm | Huygens Professional software | Scientific Volume Imaging (SVI) | - | |
| Software, algorithm | SynD | *Schmitz et al., 2011* | - | |
| Software, algorithm | DCV fusion analysis | *Moro et al., 2021* | - | The DCV fusion MATLAB script is available in GitHub at https://git.vu.nl/public-neurosciences-fga/matlab-apps/fusionanalysis2 (*Broeke, 2022*) |

## Animals

Animals were housed and bred in accordance with the Dutch and institutional guidelines. All animal experiments were approved by the animal ethical committee of the VU University/VU University Medical Centre (license number: FGA 11-03 and AVD112002017824). All animals were kept on a C57Bl/6 background. For RPH3A KO (*Schlüter et al., 1999*) and WT primary hippocampal neuron cultures, RPH3A heterozygous mice mating was timed and P1 pups were used to dissect hippocampi. Pups were genotyped prior to dissection to select RPH3A KO and WT littermates.

## Neuron culture

Primary hippocampal neurons were cultured as described before (*de Wit et al., 2009*; *Farina et al., 2015*). In short, dissected hippocampi were digested with 0.25% trypsin (Gibco) in Hanks' balanced salt solution (Sigma) with 10 mM HEPES (Life Technologies) for 20 min at 37°C. Hippocampi were washed, triturated, and counted prior to plating. For single hippocampal neurons, 1000–2000 neurons per well were plated on pre-grown micro-islands generated by plating 6000 rat glia on 18 mm glass coverslips coated with agarose and stamped with a solution of 0.1 mg/ml poly-D-lysine (Sigma) and 0.7 mg/ml rat tail collagen (BD Biosciences, *Mennerick et al., 1995*; *Wierda et al., 2007*). For continental hippocampal cultures, 20,000 neurons per well were plated on pre-grown glial plates containing 18 mm glass coverslips. All neurons were kept in neurobasal supplemented with 2% B-27, 18 mM HEPES, 0.25% Glutamax, and 0.1% Pen-Strep (all Gibco) at 37°C and 5% $CO_2$.

## Lentiviral vectors and infections

All constructs were cloned into lentiviral vectors containing a synapsin promoter to restrict expression to neurons. Lentiviral particles were produced as described before (*Naldini et al., 1996*). NPY-mCherry, NPY-pHluorin, and TeNT-IRES-mCherry were described before (*Emperador Melero et al., 2017*; *Nagai et al., 2002*). For re-expression of RPH3A in single RPH3A KO neurons, FL RPH3A was cloned into a lentiviral vector containing an IRES-NLS-mCherry to verify infection during live-cell experiments. For DCV fusion experiments at DIV14–16, neurons were infected with RPH3A-IRES-NLS-mCherry, RAB3A, and RAB27A-binding deficient RPH3A with two-point mutations (E50A and I54A, *Fukuda et al., 2004*) and SNAP25-binding deficient RPH3A (K648A, K653A, and K660A, *Ferrer-Orta et al., 2017*) at DIV1–2. The E51A/I54A double mutant of RPH3A was previously validated to lose RAB3A- and RAB27A-binding activity (*Fukuda et al., 2004*), and the K651A/K656A/K663A mutant was shown to not bind rat WT SNAP25 (*Ferrer-Orta et al., 2017*). We determined the corresponding residues to be mutated in a mouse SNAP25-binding deficient RPH3A construct (K648,653,660A). Neurons were infected with NPY-pHluorin at DIV8–9. For neurite length and co-travel experiments, FL RPH3A, truncated RPH3A (1-375), RAB3A/RAB27A-binding deficient RPH3A (*Fukuda et al., 2004*), RPH3A with two point-mutations in the C2B domain (D568N, D574N) and RPH3A phosphorylation deficient (S217A, *Foletti et al., 2001*; *Fykse et al., 1995*) were N-terminally tagged with EGFP or mCherry interspaced by a short glycine linker sequence (*Tsuboi and Fukuda, 2005*). For neurite length experiments at DIV14, all constructs were infected at DIV1–2. For co-travel experiments at DIV10 constructs were infected at DIV4.

## Immunocytochemistry

Cells were fixed with 3.7% paraformaldehyde (Merck) in phosphate-buffered saline (PBS; 137 mM NaCl, 2.7 mM KCl, 10 mM $Na_2HPO_4$, 1.8 mM $KH_2PO_4$, pH 7.4) for 12 min at room temperature (RT). Cells were immediately immunostained or kept in PBS at 4°C. For ChgB immunostainings, cells were permeabilized in 0.5% Triton X-100 (Fisher Chemical) for 10 min and blocked with 5% BSA (Acro Organic) in PBS for 30 min at RT. For all other immunostainings, cells were permeabilized with 0.5% Triton X-100 for 5 min and blocked with 2% normal goat serum (Fisher Emergo) in 0.1% Triton X-100 at RT. Cells were incubated with primary antibodies overnight at 4°C. Primary antibodies used were: polyclonal ChgB (1:500, SySy), monoclonal RPH3A (1:1000, Transduction Laboratories), polyclonal MAP2 (1:500, Abcam), polyclonal Syn1 (1:1000, #P610; 1:500, SySy), polyclonal VGLUT1 (1:500, SySy), polyclonal Homer 1 (1:500, SySy), polyclonal Tau (1:1000, SySy), monoclonal VAMP2 (1:1000, SySy), and monoclonal mCherry (1:1000, Signalway Antibody). Alexa Fluor conjugated secondary antibodies (1:1000, Invitrogen) were incubated for 1 hr at RT. Abberior secondary antibodies (1:50, Abberior) for STED imaging were incubated for 2 hr at RT. Coverslips were mounted on Mowiol (Sigma-Aldrich).

## STED imaging

STED images were acquired with a Leica SP8 STED 3× microscope with an oil immersion ×100 1.44 numerical aperture objective. Alexa Fluor 488, Abberior STAR 580, and Abberior STAR 635p were excited with 592 nm and 775 nm using a white light laser. The Abberior STAR 635p was acquired first in STED mode using 4× line accumulation, followed by Abberior STAR 580, both were depleted with 775 nm depletion laser (50% of max power). Alexa Fluor 488 was acquired in STED mode and depleted with 592 nm depletion laser (50% of max power). The Alexa Fluor 405 channel was acquired

in confocal mode. The signals were detected using a gated hybrid detector in photon-counting mode (Leica Microsystems). Z-stacks were made with a step size of 150 nm and pixel size of $22.73 \times 22.73$ nm$^2$. Finally, deconvolution was performed with Huygens Professional software (SVI).

## Live and fixed imaging

For live-cell experiments, neurons were continuously perfused in Tyrode's solution (2 mM CaCl$_2$, 2.5 mM KCl, 119 mM NaCl, 2 mM MgCl$_2$, 30 mM glucose, 25 mM HEPES; pH 7.4) at RT. For DCV fusion experiments, imaging was performed at DIV14–16 with a Zeiss AxioObserver.Z1 equipped with 561 nm and 488 nm lasers, a polychrome V, appropriate filter sets, a ×40 oil objective (NA 1.3), and an EMCCD camera (C9100-02; Hamamatsu, pixel size 200 nm). Images were acquired at 2 Hz with Axio-Vision software (version 4.8, Zeiss). Electrical stimulation was performed with two parallel platinum electrodes placed around the neuron. After 30 s of baseline, 2×8 (separated by 30 s) or 1×16 train(s) of 50 APs at 50 Hz (interspaced by 0.5 s) were initiated by a Master-8 (AMPI) and a stimulus generator (A-385, World Precision Instruments) delivering 1 ms pulses of 30 mA. NPY-pHluorin was dequenched 50 s or 80 s after the last stimulation train by superfusing Tyrode's with 50 mM NH$_4$Cl (replacing 50 mM NaCl). For a detailed protocol of DCV fusion analyses, see *Moro et al., 2021*.

For co-travel experiments, neurons were infected with NPY-mCherry or NPY-pHluorin and FL or mutant RPH3A fused to EGFP or mCherry. Imaging was performed at DIV9–10 on a Nikon Ti-E eclipse microscope with a LU4A laser system, appropriate filter sets, ×60 oil objective (NA=1.4), and EMCCD camera (Andor DU-897). Sequential images for both 488 and 561 color channels were acquired for 2 min at 2 Hz. After 2 min, both axonal and dendritic stretches were photobleached to enhance visualization of moving vesicles entering the bleached area. A galvano laser scanner was used to scan a selected area with both lasers at 100% (26.9 μs pixel dwell time). Bleaching was followed by an 8 min acquisition at 2 Hz in both channels. To visualize NPY-pHluorin, neurons were continuously perfused in Tyrode's containing 25 mM NH$_4$Cl (replacing 25 mM NaCl). NPY-pHluorin showed more resistance for bleaching. However, this did not affect the experiment as bleaching was only applied to enhance visualization of moving vesicles and facilitate analysis.

For fixed experiments, confocal images were acquired using a Zeiss LSM 510 confocal laser-scanning microscope (×40 objective; NA 1.3) and LSM 510 software (version 3.2 Zeiss) or an A1R Nikon confocal microscope with LU4A laser unit (×40 objective; NA 1.3) and NIS elements software (version 4.60, Nikon). To determine both dendrite (MAP2) and axon (Tau) length, the whole glial island was visualized by stitching 4 images ($604.7 \times 604.7$ μm$^2$).

## Analyses

For DCV fusion experiments, ImageJ was used to manually place 3×3 pixel ROIs around each NPY-pHluorin fusion event on both axons and dendrites, excluding the soma. An NPY-pHluorin event was considered a DCV fusion event if it suddenly appeared and if the maximal fluorescence was at least twice the SD above noise. Custom-written MATLAB (MathWorks, Inc) scripts were used to calculate the number and timing of fusion events. The remaining DCV number per neuron was determined as the number of fluorescent puncta during NH$_4^+$ perfusion, corrected for overlapping puncta. The released fraction was calculated by dividing the total number of fusion events by the remaining pool size. For a detailed protocol of DCV fusion analyses, see *Moro et al., 2021*.

For DCV co-travel experiments, segmented lines were drawn along the neurites and kymographs were created using the KymoResliceWide plugin in ImageJ. Puncta were considered moving if the minimal displacement during the whole 2 or 8 min acquisition was at least 3/4 μm within a 10 s period.

For fixed experiments, the number of endogenous ChgB+ puncta were counted using SynD (*Schmitz et al., 2011*; *van de Bospoort et al., 2012*) software (version 491). All puncta were divided by the mode of the first quartile of puncta intensity values (an estimate for a single DCV) to estimate the total number of DCVs per neuron. Based on this, the total number of DCVs per neuron was quantified. To determine dendritic and axonal length, a mask was created using MAP2 and Tau immunostaining, respectively. Single-pixel images were obtained from the masks. The distances between the neighboring pixels within the dendrite mask are summed together to obtain the dendrite or axon length (*Schmitz et al., 2011*). For colocalization experiments, the Pearson's and Manders' correlation coefficients were determined using the JACoP plugin (*Bolte and Cordelières, 2006*).

## Statistical analyses

Statistical tests were performed using R or GraphPad Prism. Normal distributions of all data were assessed with Shapiro-Wilk normality tests and Levene's test of homogeneity of variances. Multi-level models were used to account for variance within the animals when variance between animals significantly improved the model (*Aarts et al., 2014*). To compare two groups, unpaired Student's t-test in the case of normal distributed data or Mann-Whitney U tests for non-parametric data were used. For multiple comparisons the Kruskal-Wallis test was used for non-parametric data followed by Dunn's multiple comparisons test to compare conditions. Data is represented as boxplots (25–75% interquartile range) with the median (line), mean (+), and Tukey whiskers or bar graphs with SEM. N numbers represent number of independent experiments, with the number of individual neurons in brackets. Dots in all graphs indicate single neuron observations, unless stated otherwise.

## Acknowledgements

This work is supported by an ERC Advanced Grant (322966) of the European Union (to MV). We thank Joke Wortel for animal breeding, Robbert Zalm for cloning and producing viral particles, Desiree Schut and Lisa Laan for astrocyte culture and cell culture assistance, and Ingrid Saarloos for assistance in protein chemistry.

## Additional information

### Funding

| Funder | Grant reference number | Author |
| --- | --- | --- |
| European Research Council | 322966 | Matthijs Verhage |

The funders had no role in study design, data collection and interpretation, or the decision to submit the work for publication.

### Author contributions

Adlin Abramian, Data curation, Formal analysis, Validation, Investigation, Visualization, Writing – original draft, Project administration, Writing – review and editing; Rein I Hoogstraaten, Conceptualization, Data curation, Formal analysis, Validation, Investigation, Writing – original draft, Writing – review and editing; Fiona H Murphy, Formal analysis, Investigation, Writing – review and editing; Kathryn F McDaniel, Formal analysis, Investigation; Ruud F Toonen, Conceptualization, Supervision, Writing – review and editing; Matthijs Verhage, Conceptualization, Supervision, Funding acquisition, Writing – review and editing

### Author ORCIDs

Adlin Abramian ⓘ https://orcid.org/0009-0000-4970-2546
Fiona H Murphy ⓘ https://orcid.org/0000-0002-3995-0607
Matthijs Verhage ⓘ https://orcid.org/0000-0002-6085-7503

### Ethics

Animals were housed and bred in accordance with Dutch and institutional guidelines. All animal experiments were approved by the animal ethical committee of the VU University / VU University Medical Centre (license number: FGA 11-03 and AVD112002017824).

Joint Public Review https://doi.org/10.7554/eLife.95371.3.sa1
Author response https://doi.org/10.7554/eLife.95371.3.sa2

## Additional files

### Supplementary files
• MDAR checklist

### Data availability
Data files have been made publicly available via the DataverseNL project (https://dataverse.nl) under the DOI: https://doi.org/10.34894/CHFSZX. The DCV fusion analysis script is available in GitHub at https://git.vu.nl/public-neurosciences-fga/matlab-apps/fusionanalysis2 (*Broeke, 2022*).

The following dataset was generated:

| Author(s) | Year | Dataset title | Dataset URL | Database and Identifier |
|---|---|---|---|---|
| Verhage et al. | 2024 | Rabphilin-3A negatively regulates neuropeptide release, through its SNAP25 interaction | https://doi.org/10.34894/CHFSZX | DataverseNL, 10.34894/CHFSZX |

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
