## [Editor Report · eLife assessment]

This **important** study advances our understanding of the mechanisms of neuronal large dense-core vesicle (LDCV) secretion, which mediates neuropeptide and neurotrophin release. It describes a negative regulatory process involving the interaction of the Rab3-effector Rabphilin-3A with the SNARE fusion protein SNAP25, which limits LDCV secretion and neurite growth. The evidence in support of the authors' claims is generally **convincing**, but some conclusions, e.g regarding the role of Rabphilin-3A-controlled neurotrophin signaling in neurite growth, are **incompletely** supported. This study will be of interest to the fields of cell biology, cellular neuroscience, and neuroendocrinology.

---

## [Referee Report · Joint Public Review]

The molecular mechanisms that mediate the regulated exocytosis of neuropeptides and neurotrophins from neurons via large dense-core vesicles (LDCVs) are still incompletely understood. Motivated by their earlier discovery that the Rab3-RIM1 pathway is essential for neuronal LDCV exocytosis, the authors now examined the role of the Rab3 effector Rabphilin-3A in neuronal LDCV secretion. Based on live, confocal, and super-resolution imaging approaches, the authors provide evidence for a synaptic enrichment of Rabphilin-3A and for independent trafficking of Rabphilin-3A and LDCVs. Using an elegant NPY-pHluorin imaging approach, they show that genetic deletion of Rabphilin-3A causes an increase in electrically triggered LDCV fusion events and increased neurite length. Finally, knock-out-replacement studies, involving Rabphilin-3A mutants deficient in either Rab3- or SNAP25-binding, indicate that the synaptic enrichment of Rabphilin-3A depends on its Rab3 binding ability, while its ability to bind to SNAP25 is required for its effects on LDCV secretion and neurite development. The authors conclude that Rabphilin-3A negatively regulates LDCV exocytosis and propose that this mechanism also affects neurite growth, e.g. by limiting neurotrophin secretion. These are important findings that advance our mechanistic understanding of neuronal large dense-core vesicle (LDCV) secretion.

The major strengths of the present paper:

(i) The use of a powerful Rabphilin-3A KO mouse model.

(ii) Stringent lentiviral expression and rescue approaches as a strong genetic foundation of the study.

(iii) An elegant FRAP imaging approach.

(iv) A cutting-edge NPY-pHluorin-based imaging approach to detect LDCV fusion events.

Weaknesses of the present paper:

(i) It remains unclear why a process that affects a general synaptic SNARE fusion protein - SNAP25 - would specifically affect LDCV but not synaptic vesicle fusion.

(iii) The mechanistic links between Rabphilin-3A function, LDCV density in neurites, neurite outgrowth, and the proposed underlying mechanisms involving trophic factor release remain unresolved.

---

## [Author Response]

The following is the authors’ response to the original reviews.

**Public Reviews:**

**Joint Public Review:**
The molecular mechanisms that mediate the regulated exocytosis of neuropeptides and neurotrophins from neurons via large dense-core vesicles (LDCVs) are still incompletely understood. Motivated by their earlier discovery that the Rab3-RIM1 pathway is essential for neuronal LDCV exocytosis, the authors now examined the role of the Rab3 effector Rabphilin-3A in neuronal LDCV secretion. Based on multiple live and confocal imaging approaches, the authors provide evidence for a synaptic enrichment of Rabphilin-3A and for independent trafficking of Rabphilin-3A and LDCVs. Using an elegant NPY-pHluorin imaging approach, they show that genetic deletion of Rabphilin-3A causes an increase in electrically triggered LDCV fusion events and increased neurite length. Finally, knock-out-replacement studies, involving Rabphilin-3A mutants deficient in either Rab3- or SNAP25-binding, indicate that the synaptic enrichment of Rabphilin-3A depends on its Rab3 binding ability, while its ability to bind to SNAP25 is required for its effects on LDCV secretion and neurite development. The authors conclude that Rabphilin-3A negatively regulates LDCV exocytosis and propose that this mechanism also affects neurite growth, e.g. by limiting neurotrophin secretion. These are important findings that advance our mechanistic understanding of neuronal large dense-core vesicle (LDCV) secretion.The major strengths of the present paper are:(i) The use of a powerful Rabphilin-3A KO mouse model.(ii) Stringent lentiviral expression and rescue approaches as a strong genetic foundation of the study.(iii) An elegant FRAP imaging approach.(iv) A cutting-edge NPY-pHluorin-based imaging approach to detect LDCV fusion events.

We thank the reviewers for their positive evaluation of our manuscript.

Weaknesses that somewhat limit the convincingness of the evidence provided and the corresponding conclusions include the following:(i) The limited resolution of the various imaging approaches introduces ambiguity to several parameters (e.g. LDCV counts, definition of synaptic localization, Rabphilin-3A-LDCV colocalization, subcellular and subsynaptic localization of expressed proteins, AZ proximity of Rabphilin-3A and LDCVs) and thereby limits the reliability of corresponding conclusions. Super-resolution approaches may be required here.

We thank the reviewer for their constructive suggestion. We fully agree that super-resolution imaging would produce a more precise localization of RPH3A and co-localization with DCVs. We have now repeated our (co)-localization experiments with STED microscopy. We find that RPH3A colocalized with the pre-synaptic marker Synapsin1 and, to a lesser extent, with the post synaptic marker Homer and DCV marker chromogranin B (new Figure 1). This indicates that RPH3A is highly enriched in synapses, mostly the pre-synapse, and that RPH3A partly co-localizes with DCVs.

(ii) The description of the experimental approaches lacks detail in several places, thus complicating a stringent assessment.

We apologize for the lack of detail in explaining the experimental approaches. We have included a more detailed description in the revised manuscript.

(iii) Further analyses of the LDCV secretion data (e.g. latency, release time course) would be important in order to help pinpoint the secretory step affected by Rabphilin-3A.

We agree. To address this comment, we have now included the duration of the fusion events (new Figure S2D-F). The start time of the fusion events are shown in the cumulative plots in now Figure 3F and I. The kinetics are normal in the RPH3A KO neurons.

(iv) It remains unclear why a process that affects a general synaptic SNARE fusion protein - SNAP25 - would specifically affect LDCV but not synaptic vesicle fusion.

We agree that we have not addressed this issue systematically enough in the original manuscript. We have now added a short discussion on this topic in the Discussion of the revised manuscript (p 15, line 380-386). In brief, we do not claim full selectivity for the DCV pathway. Some effects of RPH3A deficiency on the synaptic vesicle cycle have been observed. Furthermore, because DCVs typically do not mix in the synaptic vesicle cluster and fuse outside the active zone (and outside the synapse), DCVs might be more accessible to RPH3A regulation.

(v) The mechanistic links between Rabphilin-3A function, LDCV density in neurites, neurite outgrowth, and the proposed underlying mechanisms involving trophic factor release remain unclear.

We agree that we have not addressed all these links systematically enough in the original manuscript, although we feel that we have at least postulated the best possible working model to link RPH3A function to DCV exocytosis/neurotrophic factor release and neurite outgrowth (p 15-16, line 396-400). Of course, a single study cannot support all these links with sufficient experimental evidence. We have now added a short text on what we can conclude exactly based on our experiments and how we see the links between RPH3A function, DCV exocytosis/neurotrophic factor release, neurite outgrowth and DCV density in neurites (p 13-14, line 317-325).

**Reviewer #1 (Public Review):**
Summary:The manuscript by Hoogstraaten et al. investigates the effect of constitutive Rabphilin 3A (RPH3A) ko on the exocytosis of dense core vesicles (DCV) in cultured mouse hippocampal neurons. Using mCherry- or pHluorin-tagged NPY expression and EGFP- or mCherry tagged RPHA3, the authors first analyse the colocalization of DCVs and RPH3A. Using FRAP, the authors next analyse the mobility of DCVs and RAB3A in neurites. The authors go on to determine the number of exocytotic events of DCVs in response to high-frequency electrical stimulation and find that RPH3A ko increases the number of exocytotic events by a factor 2-3, but not the fraction of released DCVs in a given cell (8x 50Hz stim). In contrast, the release fraction is also increased in RBP3A KOs when doubling the stimulation number (16x 50Hz). They further observe that RPH3A ko increases dendrite and axon length and the overall number of ChgrB-positive DCVs. However, the overall number of DCVs and dendritic length in ko cells directly correlate, indicating that the number of vesicles per dendritic length remains unaffected in the RPH3A KOs. Lentiviral co-expression of tetanus toxin (TeNT) showed a non-significant trend to reduce axon and dendrite length in RPH3a KOs. Finally, the authors use co-expression of RAB3A and SNAP25 constructs to show that RAB3A but not SNAP25 interaction is required to allow the exocytosis-enhancing effect in RPH3A KOs.While the authors' methodology is sound, the microscopy results are performed well and analyzed appropriately, but their results in larger parts do not sufficiently support their conclusions. Moreover, the experiments are not always described in sufficient detail (e.g. FRAP; DCV counts vs. neurite length) to fully understand their claims.Overall, I thus feel that the manuscript does not provide a sufficient advance in knowledge.Strengths:- The authors' methodology is sound, and the microscopy results are performed well and analyzed appropriately.- Figure 2: The exocytosis imaging is elegant and potentially very insightful. The effect in the RPH3A KOs is convincing.- Figure 4: the logic of this experiment is elegant. It shows that the increased number of DCV fusion events in RPH3A KOs is related to the interaction of RPH3A with RAB3A but not with SNAP25.

We thank the reviewer for their positive evaluation of our manuscript.

Weaknesses:- The results in larger parts do not sufficiently support the conclusions.- The experiments are not always described in sufficient detail (e.g. FRAP; DCV counts vs. neurite length) to fully understand their claims.- Not of sufficient advance in knowledge for this journal- The significance of differences in control experiments WT vs. KO varies between experiments shown in different figures.- Axons and dendrites were not analyzed separately in Figures 1 and 2.- The colocalization study in Figure 1 would require super-resolution microscopy.

To address the reviewers’ comments, we have provided a more detailed explanation of our analysis (p 19-20, line 521-542). In addition, we have repeated our colocalization experiments using STED microscopy, see Joint Public Review item (i).

**Reviewer #2 (Public Review):**
Summary:Hoogstraaten et al investigated the involvement of rabphilin-3A RPH3A in DCV fusion in neurons during calcium-triggered exocytosis at the synapse and during neurite elongation. They suggest that RPH3A acts as an inhibitory factor for LDV fusion and this is mediated partially via its interaction with SNAP25 and not Rab3A/Rab27. It is a very elegant study although several questions remain to be clarified.Strengths:The authors use state-of-the-art techniques like tracking NPY-PHluorin exocytosis and FRAP experiments to quantify these processes providing novel insight into LDCs exocytosis and the involvement of RPH3A.

We thank the reviewer for their positive evaluation of our manuscript.

Weaknesses:At the current state of the manuscript, further supportive experiments are necessary to fully support the authors' conclusions.

We thank the reviewer for their comments and suggestions. We have performed additional experiments to support our conclusions, see Joint Public Review items (i) – (iv)

**Reviewer #3 (Public Review):**
Summary:The molecular mechanism of regulated exocytosis has been extensively studied in the context of synaptic transmission. However, in addition to neurotransmitters, neurons also secrete neuropeptides and neurotrophins, which are stored in dense core vesicles (DCVs). These factors play a crucial role in cell survival, growth, and shaping the excitability of neurons. The mechanism of release for DCVs is similar, but not identical, to that used for SV exocytosis. This results in slow kinetic and low release probabilities for DCV compared to SV exocytosis. There is a limited understanding of the molecular mechanisms that underlie these differences. By investigating the role of rabphilin-3A (RPH3A), Hoogstraaten et al. uncovered for the first time a protein that inhibits DCV exocytosis in neurons.Strengths:In the current work, Hoogstraaten et al. investigate the function of rabphilin-3A (RPH3A) in DVC exocytosis. This RAB3 effector protein has been shown to possess a Ca2+ binding site and an independent SNAP25 binding site. Using colocalization analysis of confocal imaging the authors show that in hippocampal neurons RPH3A is enriched at pre- and post-synaptic sites and associates specifically with immobile DCVs. Using site-specific RPH3A mutants they found that the synaptic location was due to its RAB3 interaction site. They further could show that RPH3A inhibits DCV exocytosis due to its interaction with SNAP25. They came to that conclusion by comparing NPY-pHluorin release in WT and RPH3A KO cells and by performing rescue experiments with RPH3A mutants. Finally, the authors showed that by inhibiting stimulated DCV release, RPH3A controlled the axon and dendrite length possibly through the reduced release of neurotrophins. Thereby, they pinpoint how the proper regulation of DCV exocytosis affects neuron physiology.

We thank the reviewer for their positive evaluation of our manuscript.

Weaknesses:Data contextOne of the findings is that RPH3A accumulates at synapses and is mainly associated with immobile DCVs.However, Farina et al. (2015) showed that 66% of all DCVs are secreted at synapses and that these DCVs are immobile prior to secretion. To provide additional context to the data, it would be valuable to determine if RPH3A KO specifically enhances secretion at synapses. Additionally, the authors propose that RPH3A decreases DCV exocytosis by sequestering SNAP25 availability. At first glance, this hypothesis appears suitable. However, due to RPH3A synaptic localization, it should also limit SV exocytosis, which it does not. In this context, the only explanation for RPH3A's specific inhibition of DCV exocytosis is that RPH3A is located at a synapse site remote from the active zone, thus protecting the pool of SNAP25 involved in SV exocytosis from binding to RPH3A. This hypothesis could be tested using super-resolution microscopy.

We thank the reviewer for their suggestion. We have now performed super resolution microscopy, see Joint Public Review item (i). However, these new data do not necessarily explain the stronger effect of RP3A deficiency on DCV exocytosis, relative to SV exocytosis. We have added a short discussion on this topic to the revised manuscript, see Joint Public Review item (iv).

Technical weaknessOne technical weakness of this work consists in the proper counting of labeled DCVs. This is significant since most findings in this manuscript rely on this analysis. Since the data was acquired with epi-fluorescence or confocal microscopy, it doesn't provide the resolution to visualize individual DCVs when they are clumped. The authors use a proxy to count the number of DCVs by measuring the total fluorescence of individual large spots and dividing it by the fluorescence intensity of discrete spots assuming that these correspond to individual DCVs. This is an appropriate method but it heavily depends on the assumption that all DCVs are loaded with the same amount of NPY-pHluorin or chromogranin B (ChgB). Due to the importance of this analysis for this manuscript, I suggest that the authors show that the number of DCVs per µm2 is indeed affected by RPH3A KO using super-resolution techniques such as dSTORM, STED, SIM, or SRRF.

The reviewer is correct that this is a crucial issue, that we have not addressed optimally until now. We have previously devoted a large part of a previous manuscript to this issue, but have not referred to this previous work clearly enough. We have now clarified this (p 7, line 187-190). In brief, we have previously quantified the ratio between fluorescent intensity of ChgB and NPY-pHluorin in confocal microscopy over the number of dSTORM puncta in sparse areas of WT mouse hippocampal neurons (Persoon et al., 2018). This quantification yielded a unitary fluorescence intensity per vesicle that was very stable of different neurons. Although there might be some underestimation of the total number of DCVs when using confocal microscopy, the study of Persoon et al. (2018) has demonstrated that these parameters correlate well and that the estimations are accurate. Considering that the rF/F0 is similar in RPH3A WT and KO neurons (now Figure S2I), meaning that the intensity of NPY-pHluorin of one fusion event is comparable, we can presume that this correlation also applies for the RPH3A KO neurons.

**Recommendations for the authors:**

**Reviewer #1 (Recommendations For The Authors):**
Major points:(1) The authors perform an extensive analysis regarding the colocalization of RPH3A and DCVs (Figure 1 upper part). This analysis is hampered by the fact that the recorded data has in relation to vesicle size limited resolution (> 1 µm) to allow making strong claims here. In my view, super-resolution microscopy would be required for the co-localization studies shown in Figure 1.

We fully agree and have now performed super-resolution microscopy, see Joint Public Review item (i)

(2) The FRAP experiments (Figure 1 lower part) cannot be sufficiently understood from what is presented. The methods say that both laser channels were activated during bleaching but NPY-pHluorin is not bleached in Fig.1E. Explanation of the bleaching is not very circumspect. In 1D, it is rather EGFP-RPH3A that is entering the bleached area than the NPY vesicles. These experiments require a more careful explanation of methodology, observed results, and their interpretation. Overall, the observed effects in the original kymograph traces require a better explanation.

We acknowledge that NPY-pHluorin in Figure 1E (now Figure 2C) is not completely bleached. NPY-pHluorin appeared to be more difficult to bleach than NPY-mCherry. However, it is important to clarify that we merely bleached the neurites to remove the stationary puncta and facilitate our analysis of DCV/RPH3A dynamics. This bleaching step does not affect the interpretation of our results. We apologize that this was not clearly stated in the text and have made the necessary adjustments in legend, results- and methods section, (p 6-7, line 162-163; p 5, line 140-142 and p 19, line 508-513). Additionally, we apologize for the accidental switch of the kymographs for NPY-mCherry and EGFP-RPH3A in Figure 1D (now Figure 2B, C). We greatly appreciate identifying this error.

(3) Figure 1: The authors need to mention whether axons, dendrites, or both were analyzed throughout the different panels and how they were identified. Is it possible that axons were wrapping around dendrites in their cultures (compare e.g. Shimojo et al., 2015)? Given the limited spatial resolution and because of this wrapping, interpretation of results could be affected.

We completely agree with the reviewer’s assessment and conclusion. We are unable to distinguish axons from dendrites using this experimental design. We have made sure to specify in the text that our observation that RPH3A does not co-travel with DCVs is true for both dendrites and axons, (p 5, line 150).

(4) Figure 2: The exocytosis imaging is elegant and potentially very insightful. The effect in the RPH3A KOs is convincing. However, the authors determine the efficacy of exocytosis from NPY-pHluorin unquenching of DCVs only. This is only one of several possible parameters to read out the efficiency of exocytosis. Kinetics like e.g. delay between stimulation and start of exocytosis events or release time course of NPY after DCV fusion were not determined. Such analysis could give a better insight into what process before or after the fusion of DCVs is affected by RPH3A ko.

We fully agree with the reviewer. We have now included the duration of the fusion events (new Figure S2D-F). The start time of the fusion events are shown in the cumulative plots in now Figure 3F and I. The kinetics are normal in the RPH3A KO neurons.

Moreover, it needs to be mentioned whether 2C and D are from WT or ko cultures. It would be best to show representative examples from both genotypes.

We have now adjusted this in the new figure (now Figure 3C, D).

The number of fusion events is much increased but the release fraction is not significantly changed. While this is consistent with results in Figure 4C it is at variance with 4F. This raises questions about the reliability of the effects in RPH3A KOs.

The release fraction indicates the number of fusion events normalized to the total DCV pool. In Figure 4D, we observed a slightly bigger pool size, which explains the lack of significance when analyzing the released fraction. In Figure 4G, however, DCV pool sizes are similar between KO and WT, leading to a statistically significant effect on release fraction in KO neurons. Furthermore, Figures 4B and E distinctly show a substantial increase in fusion events in RPH3A KO neurons. This variability in pool size observed could potentially be attributed to variation in culture or inherent biological variability.

Given the increased number of ChgrB-positive DCVs in RPH3A KOs (shown in Figure 2) and that only the cumulative number of exocytosis events were analysed, how can the authors exclude that the RPH3A ko only affects vesicle number but not release, if the % change in released vesicles is not different to WT? Kinetics of release don't seem to be affected. Importantly, what was the density of NPY-pHluorin vesicles in WT vs. ko?

In Figure 2 (now Figure 5) we show that RPH3A KO neurons are larger and contain more endogenous ChgB+ puncta than WT neurons. This increased number of ChgrB+ puncta scales with their size as puncta density is not increased. A previous study (Persoon et al., 2018) has demonstrated a strong correlation between DCV number and neuron size. Our data show that RPH3A deficiency increased DCV exocytosis, but the released fraction of vesicles depends on the total number of DCVs, which we determined during live recording by dequenching NPY-pHluorin using NH4+. Considering that this is an overexpression of a heterologous DCV-fusion reporter, and not endogenous staining of DCVs, as in the case of ChgrB+ puncta, some variability is not unexpected.

Also in these experiments, the question arises of whether the authors analyse axons, dendrites, or both throughout the different panels and how they were identified.

In our experimental design we record all fusion events per cell, including both axons and dendrites but excluding the cell soma. We have clarified this in the method section, (p 19, line 508 and p 19, line 521-522).

(5) Figure 3: in D the authors show that ChgrB-pos. DCV density is slightly increased in KOs. How does this relate to the density of NPY-pHluorin DCVS in Figure 2?

We do not observe a difference in NPY-pHluorin density (see Author response image 1). However, it is important to note that we relied on tracing neurites in live recording images to determine the neuronal size. In contrast, the ChgB density was based on dendritic length using MAP2 (post-hoc) staining was limited. In addition, Chgr+ puncta represent an endogenous DCV staining, NPY-pHluorin quantification is based on overexpression of a heterologous DCV-fusion reporter. These two factors likely contribute some variability.

**Author response image 1. sa2fig1:** 

The authors show a non-significant trend of TeNT coexpression to reduce axon and dendrite lengths in RPH3A KOs. While this trend is visible, I think one cannot draw conclusions from that when not reaching significance. The argument of the authors that the increased axon and dendrite lengths are created by growth factor peptide release from DCV during culture time is interesting. However, the fact that TeNT expression shows a trend toward reducing this effect on axons/dendrites is not sufficient to prove the release of such growth factors.

We agree. We have toned down this speculation in the revised manuscript, (p 15-16, line 395-400).

Lastly, the authors don't provide insight into the mechanisms, of how RPH3A ko increases the number of DCVs per µm dendritic length in the neurons. In my view, there are too many loose ends in this story of how RPH3A ko first increases spontaneous release of DCVs and then enhances neurite growth and DCV density. Did the authors e.g. measure the spontaneous release of DCVs in their cultures?

We measured spontaneous release of DCVs during the 30s baseline recording prior to stimulation. We observed no difference in spontaneous release between WT and KO neurons (now Figure S2H). However, baseline recording lasted only 30 seconds. It is possible that this was too short to detect subtle effects.

Other points:(1) Figure 4: the logic of this experiment is elegant. It shows that the increased number of DCV fusion events in RPH3A KOs is related to the interaction of RPH3A with RAB3A but not with SNAP25. As mentioned above, it is irritating that the reduction of fusion events in KOs and on the release fraction is sometimes reaching significance, but sometimes it does not. Likewise, the absence of significant effects on DCV numbers is not consistent with the results shown in Figures 3C and D.

DCV numbers in Figure 3 (now Figure 5) are determined by staining for endogenous ChgB, whereas in Figure 4D and G DCV numbers are determined by overexpressing NPY-pHluorin and counting the dequenched puncta following a NH4+ puff.

(2) Figure 1B: truncation of the y-axis needs to be clearly indicated.

We have replaced this figure with new Figure 1 and have indicated truncations of the y-axis when needed (new Figure 1E).

(3) Page 10: "Given that neuropeptides are key modulators of adult neurogenesis (Mu et al., 2010), and that RPH3A depletion leads to increased DCV exocytosis, it is coherent that we observed longer neurites in RPH3A KO neurons." I cannot follow the argument of the authors here: what has neurogenesis to do with neurite length?

We apologize for the confusion. We have clarified this in the revised text, (p 16, line 398-400).

Minor point:There are some typos in the manuscript. e.g., page 8: "... may partially dependent on regulated secretion...; page 6: "...to dequence all...".

Thank you for noticing, we have corrected the typos.

**Reviewer #2 (Recommendations For The Authors):**
(1) Supplementary Figure S1A, in my opinion, should be in Figure 1A as it illustrates all the constructs used in this study and helps the reader to follow it up.

We thank the reviewer for their suggestion. However, we feel that with the adjustments we have made in Figure 1, the illustrations of the constructs fit better in Figure S1, since new Figure 1 shows the localization of endogenous RPH3A and not that of the constructs.

(2) One of the conclusions of the manuscript is the synaptic localization of the different RPH3A mutants. The threshold for defining synaptic localization is not clear either from the images nor from the analysis: for example, the Menders coefficient for VGut1-Syn1 which is used as a positive control, ranges from 0.65-0.95 and that of RPH3A and Syn1 ranges from 0.5-0.95. These values should be compared to all mutants and the conclusions should be based on such comparison.

We agree. We have now repeated our initial co-localization experiment with all the RPH3A mutants (now Figure S1D-F).

(3) Strengthening this figure with STED/SIM/dSTORM microscopy can verify and add a new understanding of the subtle changes of RPH3A localization.

We fully agree and have now added super-resolution microscopy data, see Joint Public Review item (i).

(4) As RAB3A/RAB27A (ΔRAB3A/RAB27A) loses the punctate distribution, please clarify how can it function at the synapse and not act as a KO. Is it sorted to the synapse and how does it is sorted to the synapse?

We used lentiviral delivery to introduce our constructs, resulting in the overexpression of ΔRAB3A/RAB27A mutant RPH3A. This overexpression likely compensates for the loss of the punctate distribution of RPH3A, thereby maintaining its limiting effect on DCV exocytosis. It is plausible that under physiological conditions, the mislocalization of RPH3A would lead to increased exocytosis, similar to what we observed in the KO.

(5) Is RPH3A expressed in both excitatory and inhibitory neurons?

We agree this is an important question. Single cell RNA-seq already suggests the protein is expressed in both, but we nevertheless decided to test expression of RPH3A protein in excitatory and inhibitory neurons, using immunocytochemistry with VGAT and VGLUT as markers in hippocampal and striatal WT neurons. We found that RPH3A is expressed in both VGLUT+ hippocampal neurons and VGAT+ striatal neurons (new Figure S1A, B).

(6) The differential use of ChgB and NPY as markers for DCVs should be clarified and compared as these are used at different stages of the manuscript.

We have previously addressed the comparison between ChgB and NPY-pHluorin (Persoon et al., 2018). We made sure to indicate this more clearly throughout the manuscript to clarify the use of the two markers.

(7) FRAP experiments- A graph describing NPY recovery should be added as a reference to 2H and discussed.

We agree. We have made the necessary adjustments (new Figure 2G).

(8) Figure 2E shows some degree of "facilitation" between the 2 8x50 pulses RPH3A KO neurons. Can the author comment on that? What was the reason for using this dual stimulation protocol?

There is indeed some facilitation between the two 8 x 50 pulses in KO neurons and to a lesser extent also in the WT neurons, which we have observed before in WT neurons (Baginska et al., 2023). Baginska et al. (2023) showed recently that different stimulation protocols can influence certain fusion dynamics, like the ratio of persistent and transient events and event duration. We used two different stimulation protocols to thoroughly investigate the effect of RPH3A on exocytosis, and assess the robustness of our findings regarding the number of fusion events. Fusion kinetics was similar in WT an KO neurons for both stimulation protocols (new Figure 2D-F).

(9) Figure 3 quantifies dendrites length and then moves to quantify both axon and dendrites for the Tetanus toxin experiment. What are the effects of KO on axon length? In the main figures, it is not mentioned but in S3 it seems not to be affected. How does it reconcile with the main conclusion on neurite length?

Figure 3H (now Figure 6C) shows the effect of the KO on axon length: the axon length is increased in RPH3A KO neurons compared to WT, similar to dendrite length. Re-expressing RPH3A in KO neurons rescues axonal length to WT levels. In Figure S3, we observe a similar trend as in main Figure 3 (new Figure 6), yet this effect did not reach significance. Based on this, we concluded that neurite length is increased upon RPH3A depletion.

(10) For lay readers, please explain the total pool and how you measured it. However, see the next comment.

We agree. We have now defined this better in the revised manuscript, (p 19, line 524-527 and p 20, line 535-539).

(11) It is a bit hard to understand if the total number of DCV was increased in the KO and if the pool size was increased and in which figure it is quantified. Some sentences like: "A trend towards a larger intracellular DCV pool in KO compared to WT neurons was observed" do not fit with "No difference in DCV pool size was observed between WT and KO neurons (Figure S2D)" or with "During stronger stimulation (16 bursts of 50 APs at 50 Hz), the total fusion and released fraction of DCVs were increased in KO neurons compared to WT". They are not directly supported, or not related to specific figures. Please indicate if the total DCVs pool, as measured by NH4, was increased and based on that, the fraction of the releasable DCVs following the long stimulation. From Figure 2H, the conclusion is an increase in fusion events. In general, NH4 is not quantified clearly- is it quantified in Figure S2C? And if it is a trend, how can it become significant in Figure 3?

We agree there has been some inconsistency in the way we describe the data on the total number of DCVs. We have addressed this in the revised text to ensure better clarity. The total DCV pool measured by NPY-pHluorin was not significantly increased in KO neurons, we see a trend towards a bigger DCV pool in the 2x8 50 Hz stimulation paradigm (now Figure S2C), therefore the released fraction of vesicles is not increased in Figure 1G (now Figure 3G). The number of DCV in Figure 3 (now Figure 5) is based on endogenous ChgB staining and not overexpression like the DCV pool measured by NPY-pHluorin. In Figure 3 (now Figure 5) we show that RPH3A KO neurons have slightly more ChgB+ puncta compared to WT.

(12) In Figure 3, the quantification is not clear, discrete puncta are not visible but rather a smear of chromogranin staining. How was it quantified? An independent method to count DCV number, size, and distribution like EM is necessary to support and add further understanding.

We acknowledge that discrete ChgB puncta are not completely visible in Figure 3 (now Figure 5). Besides the inherent limitation in resolution with confocal imaging, we believe that this is due to ChgB accumulation in the KO neurons, as shown in now Figure 5D. Nonetheless, to address this concern of the reviewer, we have selected other images that represent our dataset (now Figure 5A). Furthermore, the number of ChgB+ DCVs was calculated using SynD software (Schmitz et al., 2011; van de Bospoort et al., 2012) (see previous reply). EM would offer valuable independent confirmation on the total DCV number, size and distribution. However, with the current method we already know that vesicle numbers are at least similar. Does that justify the (major) investment in a quantitative EM study? Moreover, this issue does not affect the central message of the current study.

(13) Can the author discuss if the source of DCVs that are released at the synapse is similar or different from the source of DCVs fused while neurites elongate?

With our current experimental design, we are unable to draw conclusions regarding this aspect. We are not sure how experiments to identify this source (probably the Golgi?) would be crucial to sustain the central message of our study.

(14) An interesting and related question: what are the expression levels of RPH3A during development and neuronal growth during the nervous system development?

While we have not specifically examined the expression levels of RPH3A over development, public databases show that RPH3A expression increases over time in mice, consistent with other synaptic proteins (Blake et al., 2021; Baldarelli et al., 2021; Krupke et al., 2017). We have now added this to the revised manuscript (p 2, line 55-56).

(15) The conclusion from Figure 4 about the contribution of SNAP25 interaction to RPH3A inhibitory effect is not convincing. The data are scattered and in many neurons, high levels of fusion events were detected. Further or independent experiments are needed to support this conclusion. For example, is the interaction with SNAP25 important for its inhibitory activity in other DCV-releasing systems like adrenal medulla chromaffin cells?

We agree that further studies in other DCV-releasing systems like chromaffin cells would provide valuable insight into the role of SNAP25 interaction in RPH3A’s inhibitory effect on exocytosis. However, we believe that starting new series of experiments in another model system is outside of the scope of our current study.

(16) Furthermore, the number of DCVs in the KO is similar in this experiment, raising some more questions about the quantification of the number of vesicles, that differ, in different sections of the manuscript (points # 10,11).

The total DCV pool in the fusion experiments is measured by overexpression NPY-pHluorin, this cannot be directly compared to the number of endogenous ChgB+ DCV in Figure 3 (now Figure 5), see also item (11)

(17) The statement - "RPH3A is the only negative regulator of DCV" is not completely accurate as other DCV inhibitors like tomosyn were described before.

We agree. By this statement, we intend to convey that RPH3A is the only negative regulator of DCVs without substantial impact on synaptic vesicle exocytosis, unlike Tomosyns. We have clarified this in the revised text, (p 15, line 366-367).

(18) The support for the effect of KO on the "clustering of DCVs" is not convincing.

The intensity of endogenous ChgB puncta was decreased in RPH3A KO neurons (now Figure 5E). However, the peak intensity induced by single NPY-pHluorin labeled DCV fusion events (quanta) was unchanged (now Figure S2I). This indicates that the decrease in ChgB puncta intensity must be due to a reduced number of DCVs (quanta) in this specific location. We have interpreted that as ‘clustering’, or maybe ‘accumulation’. However, we only put forward this possibility. We are now more careful in our speculations within the text, (p 11 line 271-277).

(19) Final sentence: "where RPH3A binds available SNAP25, consequently restricting the assembly of SNARE complexes" should be either demonstrated or rephrased as no effect of trans or general SNARE complex formation is shown.

We agree. We have made the necessary adjustments in the text, (p 15, line 387-389).

(20) A scheme summarizing RPH3A's interaction with synaptic proteins and its effects on DCVs release, maybe even versus its effects on SVs release, should be considered as a figure or graphic abstract.

We have included a working model in Figure 7.

(21) Figure 4 logically should come after Figure 2 to summarize the fusion-related chapter before moving to neurite elongation.

We have placed Figure 4 after Figure 2 (now Figure 3).

**Reviewer #3 (Recommendations For The Authors):**
One important finding of this study is that RPH3A downregulates neuron size, possibly by inhibiting DCV release. Additionally, the authors demonstrated that the number of DCVs is directly proportional to the number of DCVs per µm2, and that RPH3A KO reduces DCV clustering. This conclusion was drawn by comparing ChgB with NPY-pHluorin loading of the DCVs. However, this comparison is not valid as ChgB is expressed at an endogenous level and NPY-pHluorin is over-expressed. In the KO situation where DCV exocytosis is enhanced, the available endogenous ChgB may be depleted faster than the overexpressed NPY-pHluorin. Hoogstraaten et al. should either perform a study in which ChgB is overexpressed to test whether the difference in DCV remains or at least provides an alternative interpretation of their data.

We thank the reviewer for this comment. The reviewer challenges one or two conclusions in our original manuscript (It is not entirely clear to what exactly “This conclusion” refers): (a) “the number of DCVs is directly proportional to the number of DCVs per µm2”, and (b) “that RPH3A KO reduces DCV clustering”. The reviewer probably means that the number of DCVs per neuron is directly proportional to size of the neuron (a) and states this (these) conclusion(s) are “not valid as ChgB is expressed at an endogenous level and NPY-pHluorin is over-expressed” because “endogenous ChgB may be depleted faster than the overexpressed NPY-pHluorin”. We have three arguments to conclude that faster depletion of ChgB cannot affect these two conclusions: (1) DCVs bud off from the Golgi with newly synthesized (fresh) ChgB. Whether or not a larger fraction of DCVs is released does not influence this initial ChgB loading into DCVs (together with over-expressed NPY-pHluorin); (2) in hippocampal neurons merely 1-6% of the total DCV pool undergoes exocytosis (the current study and also extensively demonstrated in Persoon et al., 2018). RPH3A KO neurons release few percent more of the total DCV pool. Hence, “depletion of ChgB” is only marginally different between experimental groups; and (c) the proposed experiment overexpressing ChgB will not help scrutinize our current conclusions as ChgB overexpression is known to affect DCV biogenesis and the total DCV pool, most likely much more than a few percent more release by RPH3A deficiency.

Hoogstraaten et al. conducted a thorough analysis of the impact of RPH3A KO and its rescue using various mutants on dendrite and axon length (see Supplementary Figure 3). However, they did not test the effect of the ΔSNAP25 mutant. The authors demonstrated that this mutant is the least efficient in rescuing DCV exocytosis (Figure 4E). Hence the neurons expressing this mutant should have a similar size to the KO neurons. This finding would strongly support the argument that DCV exocytosis regulates neuron size. Otherwise, it would suggest that RPH3A may have a function in regulating exocytosis at the growth cones that is independent of SNAP25. Since the authors most probably have the data that allows them to measure the neuron size (acquired for Supplementary Figure 2), I suggest that they perform the required analysis.

We agree this is important and performed new experiments to determine the dendrite length of RPH3A WT, KO and KO neurons expressing the ΔSNAP25 mutant. We observed that the dendrite length of RPH3A KO neurons expressing ΔSNAP25 mutant is indeed similar to KO neurons (new Figure S3C). Although not significant we observe a clear trend towards bigger neurons compared to WT. This strengthens our conclusion that increased DCV exocytosis contributes to the observed increased neuronal size.

The authors displayed the result of DCV exocytosis in two ways. One is by showing the number of exocytosis events the other is to display the proportion of DCVs that were secreted. They do the latter by dividing the secreted DCV by the total number of DCVs. These are visualized at the end of the experiment through NH4+ application. While this method works well for synaptic secretion as the marker of SV is localized to the SV membrane and remains at the synapse upon SV exocytosis, it cannot be applied in the same manner when it is the DCV content that is labeled as it is released upon secretion. Hence, the total pool of vesicles should be the number of DCV counted upon NH4+ application in addition to those that are secreted. This way of analyzing the total pool of DCV might also explain the difference in this pool size between KO neurons stimulated two times with 8 stimuli instead of one time with 16 stimuli (Sup Fig 2 C and D). This is an important point as it affects the conclusions drawn from Figure 2.

We thank the reviewed for this comment. We agree, and we have made the necessary adjustments throughout the manuscript.

The kymogram of DCV exocytic events displayed in Figure 2D shows a majority of persistent (>20s long) events. This is strange as NPY-pHluori corresponds to the released cargo. Previous work using the same labeling and stimulation technique showed that content release occurs in less than 10s (Baginska et al. 2023). The authors should comment on that difference.

In Baginska et al. (2023), the authors distinguished between persistent and transient events. The transient events are shorter than 10s for the 2x8 and 16x stimulation paradigms, whereas persistent events can last for more than 10s. In our study we did not make this distinction. However, in response to this reviewer, we have now quantified the fusion duration per cell. These new data show that the mean duration is similar between genotypes for both stimulation paradigms. We have added these new data (new Figure S2D-F).

In Figures 1D and E, some puncta in the kymogram appeared to persist after bleaching. This raises questions about the effectiveness of the bleaching procedure for the FRAP experiment.

The reviewer is correct that NPY-pHluorin in Figure 1E (now Figure 2C) is not fully bleached. NPY-pHluorin was more resistant to bleaching than NPY-mCherry. However, we merely bleached the neurites to facilitate our analysis by reducing fluorescence of the stationary puncta without causing phototoxicity. Some remaining fluorescence after bleaching does not affect our conclusions in any way.

In the discussion, the paragraph titled "RPH3A does not travel with DCVs in hippocampal neurons" is quite confusing and would benefit from a streamlined explanation.

We thank the reviewed for this comment. We made the necessary adjustments to make this paragraph clearer, (p 14, line 339-351).

First paragraph of page 8 "TeNT expression in KO neurons restored neurite length to WT levels. When compared to KO neurons without TeNT, neurite length was not significantly decreased but displayed a trend towards WT levels (Figure 3G, H)." These two sentences are confusing as they seem contradictory.

We agree that this conclusion has been too strong. However, we do not see a contradiction. The significant effect between KO and control neurons on both axon and dendrite length is lost upon TeNT expression (which forms the basis for our conclusions cited by the reviewer, now Figure 6B, C). While the difference between KO neurons +/- TeNT did not reach statistical significance. The (strong) trend is clearly in the same direction. We have refined our original conclusion in the revised manuscript, (p 12, line 304-306).

The data availability statement is missing.

We have added the data availability statement, (p 21, line 571-572).